# KNOWPLAN: Knowledge-Driven AI Agents for Smart Degree Pathway Planning

## Abstract

Abstract Recent advances in large language models (LLMs) provide powerful capabilities for knowledge-driven course planning. However, building reliable, constraint- aware study planners from publicly available course webpages remains challenging due to heterogeneous data sources, complex multi-logic prerequisites, and multi-requirement constraints. To address these challenges, this paper proposes KNOWPLAN, a proactive, self-evolving multi-agent AI platform that integrates LLM-based extraction, knowledge-graph construction, and constraint-aware reasoning to generate adaptive, personalized study plans. This platform brings together scientific inquiry, technical challenges, and practical utility within a coherent and unified framework. The scientific inquiry focuses on two fundamental problems: the heterogeneity of publicly available university catalog webpages, and the limitations of traditional graph structures in handling prerequisite logic. The technical challenges include extracting structured course information from diverse catalogs, modeling prerequisite structures as hypergraphs, and extracting critical paths under multi-dependency conditions. To tackle these issues, we propose a multi-LLM-driven Agent Forest to handle webpage heterogeneity, introduce the Logic Adjacency Matrix as a novel representation of course prerequisite graphs, and develop the Multi-Dependency Critical Path Extraction algorithm to support effective course planning. These components represent the core technical highlights of this work. On the engineering side, a major contribution of this work is the design of a modular end-to-end pipeline composed of four key components: the Agent Forest, Graph-Construction Agent, Course Planning Agent, and Curriculum Alignment Agent. LLMs are integrated at various stages of this pipeline to support course information extraction, prerequisite cycle resolution, personalized course recommendation, and term-level schedule generation tailored to individual preferences and academic backgrounds. Across multiple universities, KNOWPLAN achieves 99.5% accuracy on major requirements and 98.7% on prerequisites. By combining graph-based reasoning with a term-level scheduler, it generates feasible and personalized study plans that respect preferences, workload limits, and policy exceptions, outperforming state-of-the-art methods.

## 1 Introduction

Academic degree pathway planning is especially important for students as it provides a clear and structured roadmap through their academic journey. A well-designed pathway helps them sequence courses correctly, balance academic workloads, and avoid the risk of taking unnecessary credits that waste both time and financial resources. It also ensures that students remain aligned with their long-term goals, whether pursuing graduate studies, professional licensure, or career opportunities. By reducing uncertainty, minimizing delays, and promoting efficient progress toward graduation, pathway planning becomes an essential tool for supporting undergraduate success and improving overall educational outcomes.

With the rapid advancement of natural language processing (NLP), particularly through large language models (LLMs) (OpenAI et al., 2024), numerous relevant studies have emerged in recent years (Ng & Fung, 2024; Spahic-Bogdanovic et al., 2025; Chun et al., 2025). Among these approaches, one of the most significant technological innovations in education is the application of Knowledge Graphs (KGs), which represent an interconnected web of information, concepts, and relationships

carefully structured to capture the diverse dimensions of educational content and pedagogical practices (Hogan et al., 2021; Abu-Salih & Alotaibi, 2024). Building on course-based KGs, various personalized recommendation systems have been developed to support both educational institutions and individual learners. Some of these systems serve general purposes, while others are tailored to academic preferences or designed to address job-market demands. Despite these advances, several challenges remain: (1) diverse webpage structures, (2) heterogeneous data integration, (3) complex prerequisite modeling, (4) balancing personalization with institutional rules, (5) multi-constraint and conflicts, 6) semester-level course mapping, and (7) real-time data updates and maintenance.

However, relying on heterogeneous public data introduces significant technical challenges that are both academically interesting and technically demanding: (1) **structural heterogeneity**: how to robustly handle diverse HTML/CSS/JavaScript structures across 6,072+ institutions and robustly parse unstructured natural-language catalog; (2) **prerequisite representation with hyper-graph**: how to use hyper-graph to represent both the connections and the OR–AND logic present in course prerequisites(Gao et al., 2022); (3) **adjacency matrix limitations**: traditional adjacency matrices, serving as the foundational data structure for many graph representations and algorithms, fail when prerequisite structures contain OR-AND logic; (4) **Critial path extraction issue from hyper-graph**: how to extract the critical course path from hyper-graph since the dysfunction of adjacency matrix introduces a cascade of derivative issues in graph methods, particularly evident in algorithms such as the critical path method, which assumes linear or singular path dependencies, while university course planning involves multiple, nested, and conditional path dependencies that violate these assumptions; (5) **cycle resolution**: providing a more principled cycle-handling strategy than simple DFS-style heuristics via LLM-based semantic disambiguation; (6) **multi-constraint unification**: how to incorporate competing objectives (early graduation, cost reduction, career alignment, personalized GE) into a unified planning framework; (7) **policy compliance**: ensuring constraint satisfaction without institutional rule engines. These challenges, arising from the structural complexity of course catalog data and the limitations of traditional graph representations, highlight the need for new methods or algorithmic frameworks to overcome the resulting methodological limitations.

To address these challenges, we propose a multi-LLM-driven Agent Forest to manage webpage heterogeneity, introduce the Logic Adjacency Matrix as a novel representation for course-prerequisite graphs, and develop the Multi-Dependency Critical Path Extraction algorithm to facilitate effective course planning. Together, these components constitute the central technical contributions of this work. From an engineering perspective, a major highlight lies in the design of a modular, end-to-end pipeline comprising four key modules: the Agent Forest, the Graph-Construction Agent, the Course Planning Agent, and the Curriculum Alignment Agent. LLMs are integrated throughout this pipeline to enable course-information extraction, prerequisite-cycle resolution, personalized course recommendations, and term-level schedule generation tailored to individual preferences and academic backgrounds.

## 2 RELATED WORK

Recent studies in educational AI have facilitated in personalized academic planning and course recommendation. We review this literature in two strands: (i) systems that use large language models (LLMs) to generate personalized learning paths (Ng & Fung, 2024), and (ii) methods that combine knowledge graphs (KGs) with retrieval-augmented generation (RAG) (Lewis et al., 2021) to ground LLM reasoning and control hallucination.

### 2.1 LEARNING PLANNING WITH LLMS

Early work on academic pathway planning with LLMs treats course planning as a natural language generation task. Ng and Fung's *Educational Personalized Learning Path Planning* (PLPP) shows how crafted prompts can guide GPT-4 and LLama-2 to produce coherent, learner-specific paths; experiments demonstrate improved accuracy, user satisfaction, and pedagogical quality relative to baseline text-only methods (Ng & Fung, 2024). This approach highlights the promise of LLMs but also relies heavily on prompt engineering to encode prerequisite logic and does not enforce feasibility constraints explicitly. Building on this, *PlanGlow* introduces an LLM-driven system that generates personalized study plans with transparent explanations and user-adjustable parameters

(Chun et al., 2025) Through mixed-methods evaluation, PlanGlow improves usability, explainability, and controllability over a GPT-4 baseline and Khan Academy's Khanmigo tutor(Chun et al., 2025)

Another research integrates LLMs with structured data and multi-agent architectures. *iModuleBuddy* is a hybrid AI-based academic planning system that combines vector-based retrieval and course descriptions (Spahic-Bogdanovic et al., 2025). It uses a JobRanking algorithm (Boonchob et al., 2022) to align students' professional experience with course relevance and a multi-agent planner to assemble multi-semester study plans, providing career-focused and preference-based variants (Spahic-Bogdanovic et al., 2025). A knowledge graph links occupations, competencies, and courses, enabling retrieval-augmented generation and reducing cold-start problems (Spahic-Bogdanovic et al., 2025). Despite these innovations, the system is still under development, and it does not yet integrate fine-grained prerequisite or scheduling constraints.

## 2.2 KG AND RAG BASED METHODS

To address the uncertainty of LLM reasoning and mitigate hallucinations (Shojaee* et al., 2025), recent methods couple LLMs with graph through retrieval-augmented generation (RAG) (Han et al., 2025). The KARMA framework leverages a multi-agent architecture for knowledge graph extraction and enrichment, incorporating cross-agent verification to improve reliability (Lu & Wang, 2025). Zhu et al. further propose KG$^2$RAG, a KG-guided RAG framework that uses semantic retrieval to obtain seed chunks and then expands and organizes them using KG topology, thereby providing fact-level relationships between passages and improving retrieval quality on benchmarks such as HotpotQA (Zhu et al., 2025). Building further, Jiang et al.'s *KG-Agent* integrates an LLM with a toolbox, a KG-based executor, and a memory module. This autonomous agent iteratively selects tools and updates its knowledge memory to perform multi-hop reasoning, enabling a smaller LLaMA-7B model to outperform larger LLMs on complex KG-QA tasks (Jiang et al., 2024).

Platform like iModuleBuddy already leverage KGs to link courses to occupational competencies and use RAG to generate recommendations (Spahic-Bogdanovic et al., 2025). However, most educational RAG systems still treat constraints loosely and do not integrate exact solvers for timetable feasibility. Our work takes a step further by proposing agent forest to harmonize heterogeneous catalog data into a KG, using Graph-RAG to retrieve prerequisite and policy information, and embedding these within a constraint programming solver to ensure that generated plans are both pedagogically sound and schedule-feasible.

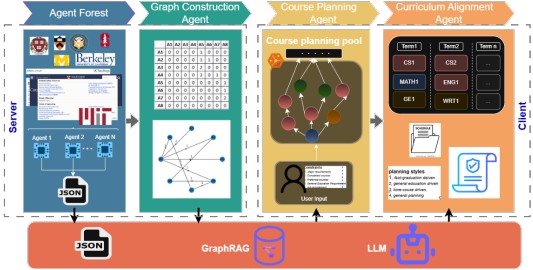

Figure 1: Pipeline of KNOWPLAN. Workflow: (1) GPT-5 + Llama3.3-70B + Qwen3-32B with few-shot prompting retrieves catalog and requirement webpages; (2) LLMs with zero-shot prompting extract required courses and logical relationships; (3) an HTML parser and LLMs extract courses, prerequisites, and descriptions; (4) results are stored in JSON files and indexed in Graph-RAG. Prompt designs are provided in the Supplementary Materials.

## 3 METHOD

We propose **KNOWPLAN**, a multi-agent AI platform leveraging LLMs, knowledge graphs, and RAG for degree pathway planning. The overall architecture is illustrated in Figure 5.

## 3.1 AGENT FOREST FOR PARSING HETEROGENEOUS WEBPAGES

Course catalogs are publicly available but exhibit heterogeneous webpage structures across institutions. Each university exhibits unique HTML structures, CSS layouts, JavaScript interactions, and data organization patterns. A single-agent approach fails; traditional multi-agent systems (2–3 agents) cannot adapt to thousands of unique structures. We introduce an **Agent Forest**, where **each institution gets a dedicated agent with tailored prompts**, enabling systematic handling of extreme heterogeneity at scale. We define and empirically evaluate an **Agent Forest architecture** with dedicated agents per institution. **Institution-specific agent architecture.** Each institution maintains its own Agent Forest configuration with institution-specific prompts tailored to its unique website structure, catalog organization, and term system. Agents are instantiated from shared prompt patterns, but each university keeps its own agent definition. Manual correction rates are 2–5% for standard HTML catalogs and 8–12% for non-standard/JavaScript-heavy pages (Sec. 4.1). Agent Forest extraction is embarrassingly parallel across institutions (Sec. **??**). The basic workflow of the algorithm is as follows:

**Agent motivations.** Each agent addresses a specific research problem: (i) *Agent Forest* handles extreme heterogeneity across 6,072+ institutions; (ii) *Graph-Construction Agent* transforms unstructured data into structured knowledge graphs, resolving circular dependencies; (iii) *Course Planning Agent* performs multi-objective optimization balancing major/GE requirements, interests, difficulty, and time-to-degree; (iv) *Curriculum Alignment Agent* refines coarse plans into detailed, term-assigned, conflict-free schedules. See Supplement Section **??** for step-by-step examples.

We employ a collaborative, multi-LLM approach (Darwish et al., 2025; Wu et al., 2023) with *Llama3.3-70B*, *Qwen3-32B*, and *GPT5*. Each LLM independently extracts information; GPT-5 arbitrates conflicts. Unstructured data are transformed into uniform representations stored in JSON files in GraphRAG tables (catalog webpage, requirement webpage, requirement rules, required courses, course information, course dependency logic).

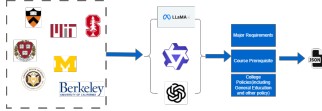

Figure 2: GPT-5 + Llama3.3-70B + Qwen3-32B with few-shot prompting retrieves catalog and requirement webpages.

## 3.2 GRAPH-CONSTRUCTION AGENT

Graph-Construction Agent creates a course KG capturing prerequisite relationships from Agent Forest's unified representations. The KG is a complex, directed, multi-connected graph where reliability directly affects downstream accuracy. Key challenges: handling logical expressions of prerequisites with multiple options and addressing circular dependencies. Most course prerequisite structures can be expressed as OR–AND logic expressions. For example, the expression "B1: (A1 or A2) and (A3 or A4)" means that course B1 has two prerequisite slots: one course chosen from {A1, A2} and one course chosen from {A3, A4}, where B1, A1, A2, A3, and A4 denote five distinct courses. This expression contains two AND blocks, namely $(A_1 or A_2)$ and $(A_3 or A_4)$.

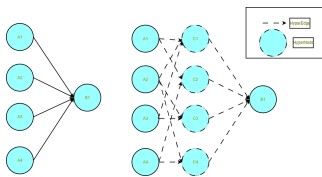

Figure 3: hypernode and hyperedge challenge,(1) tradition nodes and edges on the left, and (2) combination of traditional nodes and hypernodes/edges on the right.

In a traditional graph, it is almost impossible to encode such complex logical relations using only nodes and edges, because each node is treated as an independent element. However, a course node

naturally has two roles: it is both an independent entity (a single course) and a member of a prerequisite group (one option within an OR block). Representing this dual role, and more generally, representing complex OR–AND logic on graphs, poses a significant challenge for graph theory. A common solution is to use hypergraphs, which introduce hypernodes to encode different logical relations (see Figure 3.1 and Figure 3.2, where four hypernodes are needed to represent the prerequisites of B1). The advantage is that many traditional graph toolkits and algorithms can still be applied. However, as the number of courses and prerequisite groups grows, the number of hypernodes can increase exponentially, making downstream operations such as critical-path and shortest-path detection intractable. This motivates our central question: can we preserve all prerequisite logic while still working within a traditional graph representation of the course knowledge graph?

To address this challenge, we propose a Logic Adjacency Matrix (LAM) based on OR–AND logic expressions, defined as follows:

$$G = (V, E), \qquad A = (a_{mn}).$$

$$a_{mn} = \begin{cases} i, & \text{if } (v_m, v_n) \in E \text{ and } \mathrm{AND}(v_m) = i, \\ 0, & \text{otherwise.} \end{cases}$$

$\mathrm{AND}(v_m)$ is the index of the AND cell in the prerequisites. For example, in

$$B_1 = (A_1 \vee A_2) \wedge (A_3 \vee A_4),$$

there are two AND cells, $(A_1 \vee A_2)$ and $(A_3 \vee A_4)$, and all nonzero entries in the column for $B_1$ that correspond to the first cell share the same value $i = 1$, while those corresponding to the second cell share $i = 2$.

For each column in the Logic Adjacency Matrix (LAM), entries with the same nonzero value represent an OR relationship, while distinct nonzero values within that column represent AND relationships between different prerequisite groups. It is easy to verify that the LAM is logically equivalent to the original prerequisite specification: it preserves both connectivity and OR–AND structure while remaining a standard adjacency matrix on the underlying course graph, thus providing a strong tool for course knowledge-graph construction.

Circular dependencies create cycles, violating the acyclic structure. We use DFS cycle detection and LLM-based semantic disambiguation to resolve cycles. **Why LLM-based cycle resolution?** DFS cycle detection cannot distinguish true cycles from apparent cycles caused by ambiguous catalog language. For example, a naive parser may produce a mutual cycle between Course 1A and Course 1B. Our RAG-enhanced LLM reads catalog snippets (e.g., "Course 1A–1B may be taken in either order") and correctly concludes they are peers, not prerequisites. The LLM resolves semantic ambiguities, augmenting algorithmic cycle detection with contextual understanding.

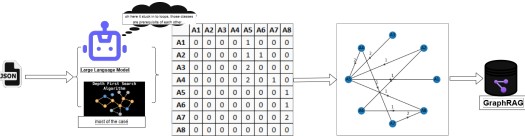

Figure 4: Workflow of Graph Construction Agent

### 3.2.1 COURSE PLANNING AGENT

Course planning is a task of mining critical paths from the curricular knowledge graph. Since LAM becomes more complex than traditional adjacency matrix, the classic critical path detection method dysfunctions for course planning due to the logic and multi prerequisites. Due to its key role in this taskl, we proposed a Multi-Dependency Critical Path Detection Algorithm for this challenge. Its pseudo code is given in Supplementary matrial. With Multi-Dependency Critical Path Detection AlgorithmCourse Planning Agent fertilizes LLM to produces personalized coarse planning options under multiple constraints: major requirements, completed courses, preferred courses, GE requirements, and job requirements. Required courses ensure prerequisites are completed; completed

courses block re-recommendation; preferred courses and job targets induce preference priors; GE buckets define coverage gaps; unit bounds and difficulty preferences act as selection priors. Output is dependency-consistent coarse plan pools covering major + GE requirements.

**Constraint handling details.** Major requirements are hard constraints; prerequisites are added via graph traversal. Completed courses are marked satisfied. Preferred courses and job requirements are soft constraints via preference prior $\pi_{\text{int}}(v)$. GE requirements use gap analysis to identify unmet buckets. Unit bounds and difficulty preferences act as regularization terms.

We form a planning subgraph from major requirement roots and completed courses ($\mathcal{C}_{\text{done}}$, $\mathcal{I}_{\text{interest}}$) and rank candidates with:

$$s_{\text{node}}(v) = \phi_{\text{prereq}}(v)\Big(\lambda_1\, \mathbf{r}(v) + \lambda_2\, \text{BM25}(v, q) + \lambda_3\, \pi(v)\Big), \tag{1}$$

where $\phi_{\text{prereq}}(v) \in \{0, 1\}$ enforces prerequisite feasibility; $\mathbf{r}(v)$ is graph-proximity to requirement roots; $\text{BM25}(v, q)$ adds text relevance; $\pi(v) = \pi_{\text{int}}(v) + \beta'\, \text{gap}_{\text{GE}}(v, \mathcal{G}_{\text{GE}})$ measures interest alignment and GE gap filling. **Educational theory grounding:** Prerequisite feasibility reflects scaffolded learning (Bloom's taxonomy) and Zone of Proximal Development (Vygotsky). Graph proximity models constructivist learning; interest alignment implements student-centered learning (Dewey, 1938) and self-determination theory (Deci & Ryan, 1985). GE gap term implements curriculum alignment theory (Tyler, 1949).

From the subgraph, we choose courses with $x_v \in \{0, 1\}$ and maximize:

$$\mathcal{J}(\mathcal{P}) = \sum_v s_{\text{node}}(v)\, x_v + \mu_1\, \text{Cover}_{\text{major}}(\mathcal{P}) + \mu_2\, \text{Cover}_{\text{GE}}(\mathcal{P}) - \nu_1\, \text{Viol}_{\text{prereq}}(\mathcal{P}) - \nu_2\, |\text{Units}(\mathcal{P}) - U^\star| - \nu_3\, \text{DiffGap}(\mathcal{P}; d^\star) \tag{2}$$

**Educational theory grounding:** Major/GE coverage encodes competency-based education. Prerequisite violation penalty and unit constraint motivated by Cognitive Load Theory (Sweller, 1988) and Spaced Repetition (Ebbinghaus, 1885). Difficulty gap implements progressive difficulty. Tunable weights and planning modes reflect adaptive learning pathways. subject to prerequisite closure: $x_v \leq x_u \quad \forall (u \xrightarrow{\text{IS\_PREREQ}} v)$. We produce top-$K$ coarse plan pools $\{\mathcal{P}^{(k)}\}_{k=1}^K$ via CP-SAT enumeration, yielding term-agnostic, dependency-consistent pools covering major/GE requirements. (Term assignment by Curriculum Alignment Agent in Sec. 3.2.2.)

**Top-$K$ search strategy and LLM overhead mitigation.** Top-$K$ enumeration uses CP-SAT with iterative no-good cuts. LLM calls limited to: (i) Graph-RAG subgraph retrieval (one call per query) and (ii) preference-to-weight mapping (one call per planning mode). CP-SAT handles all optimization deterministically. See Supplement Sec. **??** for evaluation.

**Trade-off resolution.** When objectives conflict, top-$K$ provides multiple Pareto-optimal solutions. Each plan $\mathcal{P}^{(k)}$ represents a distinct trade-off. The Curriculum Alignment Agent refines trade-offs by ranking schedules under four modes, each with distinct weight vectors.

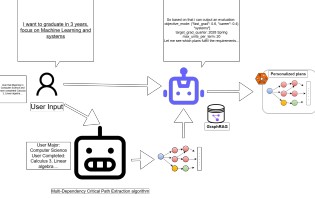

Figure 5: Workflow of the Course Planning Agent.

### 3.2.2 CURRICULUM ALIGNMENT AGENT WITH REAL-TIME PLANNING

Curriculum Alignment Agent refines coarse plan pools into detailed, term-assigned schedules. Given pools $\{\mathcal{P}^{(k)}\}_{k=1}^K$, it (i) tests schedulability against real offerings and policies and (ii) assigns courses to concrete terms/sections via CP-SAT scheduler. The agent couples LLM-assisted preference/policy parsing with exact CP-SAT scheduling. Four target-driven modes: fast-graduation (0.8, 0.1, 0.1), career-driven (0.4, 0.5, 0.1), course-driven (0.3, 0.6, 0.1), discovery (0.3, 0.2, 0.5) with

weight vector $\boldsymbol{\lambda} = (\lambda_{\text{deferral}}, \lambda_{\text{pref}}, \lambda_{\text{smooth}})$ (Table 1). Output is ranked conflict-free schedules with concrete assignments. **Scheduling model: CP-SAT formulation.** We use a standard CP-SAT integer programming model; the novel contribution is the principled interface between LLM-parsed policies/preferences and this solver. For each pool $\mathcal{P}^{(k)}$, we instantiate binary decision variables $x_{c,s,t} \in \{0,1\}$ indicating whether course $c$ is taken in section $s$ during term $t$. Hard constraints include: (i) offering availability, (ii) non-overlap of section time blocks, (iii) per-term unit brackets, (iv) prerequisite/co-requisite timing, (v) institution-specific limits (repeat/credit rules). The CP-SAT solver returns a feasibility flag and concrete assignment.

**Hyperparameters: principled LLM-to-solver interface.** The LLM sets weights and soft-constraint flags via weight vector $\boldsymbol{\lambda} = (\lambda_{\text{deferral}}, \lambda_{\text{pref}}, \lambda_{\text{smooth}})$ for: (a) deferral minimization, (b) preference fit, (c) workload smoothing. The four modes are different fixed choices of $\boldsymbol{\lambda}$ plus soft constraint toggling. The LLM maps free-text input into $\boldsymbol{\lambda}$ and preference flags. Once $\boldsymbol{\lambda}$ is chosen, CP-SAT minimizes a weighted sum subject to hard constraints. The solver remains fully deterministic and exact.

**Weight selection.** Weights were manually selected based on domain knowledge and desired trade-offs. We do not claim these are globally optimal; they represent one effective configuration validated on 330 scenarios (100% schedulability, 0 violations, Supplement Sec. **??**) and shown robust ($\pm 2\%$ quality under $\pm 50\%$ perturbations, Supplement Sec. **??**). Alternative weight choices may achieve similar performance. Table 1 summarizes configurations and operational semantics.

Table 1: Weight configurations $\boldsymbol{\lambda} = (\lambda_{\text{deferral}}, \lambda_{\text{pref}}, \lambda_{\text{smooth}})$ for target-driven scheduling modes. GE = General Education.

| Mode | $\lambda_{\text{deferral}}$ | $\lambda_{\text{pref}}$ | $\lambda_{\text{smooth}}$ |
|---|---|---|---|
| Fast-graduation (default) | 0.8 | 0.1 | 0.1 |
| Career-driven | 0.4 | 0.5 | 0.1 |
| Course-driven | 0.3 | 0.6 | 0.1 |
| Discovery | 0.3 | 0.2 | 0.5 |

**Operational semantics.** Fast-graduation: 10.7 terms mean time-to-degree (vs. 12.4 discovery). Career-driven: 85% career-relevance (vs. 62% fast-graduation). Course-driven: 92% preference satisfaction (vs. 68% fast-graduation). Discovery: workload variance 2.3 units (vs. 5.7 fast-graduation).

**Policy formalization.** LLM produces structured entries (JSON) from policy text (e.g., unit caps, repeat limits, co-requisites), compiled deterministically into CP-SAT constraints. Rule-based compilation, no LLM reasoning at solve time.

**Selection and ranking.** We solve per $\mathcal{P}^{(k)}$, collect feasible schedules, and rank them by the chosen objective; infeasible pools are returned with succinct, constraint-grounded explanations (e.g., "not offered," "time conflict," "unit cap exceeded") drafted by the LLM.

**LLM accuracy impact.** Errors propagate through subgraph retrieval (mitigated by multi-hop expansion), preference interpretation (shifts rankings, not feasibility), and policy formalization (96.2% precision ensures compliance). CP-SAT guarantees consistency; top-$K$ provides fallbacks.

## 4 EXPERIMENTS

### 4.1 DATA COLLECTION AND PREPARATION

**Data collection methodology.** We collect institutional data through a two-stage process: (1) obtain U.S. postsecondary institutions from IPEDS (NCES), including website URLs (6,072+ institutions); (2) Agent Forest (Sec. 3.1) crawls each institution's catalog and degree-requirement pages, retrieves and stores HTML content, then performs automated extraction. Dataset: https://nces.ed.gov/ipeds/datacenter/DataFiles.aspx?gotoReportId=7&fromIpeds=true&sid=fb9d64a3-7797-4a2a-a9a9-cc5918597e08&rtid=7.

Table 2: Quantitative coverage across four universities (Agent Forest).

| University | Category | KNOWPLAN | Ground Truth | University | Category | KNOWPLAN | Ground Truth |
|---|---|---|---|---|---|---|---|
| **University 1** | department | 85 | 87 | **University 2** | department | 15 | 15 |
| | major | 167 | 167 | | major | 85 | 85 |
| | courses | 7253 | 7380 | | courses | 5409 | 5722 |
| **University 3** | department | 65 | 66 | **University 4** | department | 77 | 77 |
| | major | 106 | 108 | | major | 84 | 84 |
| | courses | 10194 | 10194 | | courses | 15380 | 15384 |

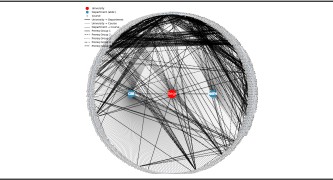 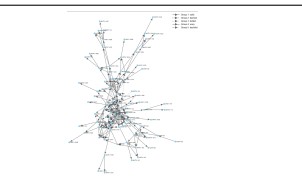 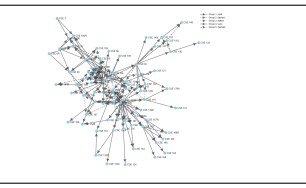

Figure 6: Structural overviews for University 1. (a) Radial map. (b) Mathematics department graph. (c) Computer Science and Engineering department graph.

**Ground-truth dataset.** (i) *Institution-level:* Manually annotated departments, majors, and courses (3,132 courses, 4 universities, Table 2), cross-checked by two experts. Evaluates Agent Forest extraction accuracy; does not include prerequisite relationships. (ii) *Prerequisites:* Two domain experts independently annotated 1,248 directed prerequisite edges from ≈460 courses (12 majors, 4 universities, 2024–2025 catalogs), using shared rubric. Cohen's $\kappa$=0.92. Gold-standard: 98.7% precision, 97.9% recall, 98.3% $F_1$ (Supplement Sec. **??**, Table **??**). (iii) *Policy rules:* Two compliance specialists independently mapped 240 policy snippets (2023–2025 catalogs) into normalized JSON constraint schema. Cohen's $\kappa$=0.89. 96.2% precision, 94.7% recall (Supplement Sec. **??**). **Manual correction rates:** 2–5% (standard HTML), 8–12% (non-standard/JavaScript-heavy). Maintenance scales linearly (sec:scalability-cost).

### 4.2 RESULTS

#### 4.2.1 AGENT FOREST

**Extraction accuracy and coverage.** Agent Forest extracts course information from diverse institutional websites. Across four institutions: 95.2–98.7% precision, 93.8–97.9% recall. Standard HTML: 97–99% accuracy; JavaScript-heavy: 8–12% manual correction. Institution-specific prompt adaptation enables accurate extraction. **Institution-specific adaptation analysis:** Single universal agent (72–78% precision), small multi-agent (85–91%), Agent Forest (95–99%). Agent Forest shows 12–15 percentage point gains for non-standard structures.

Agent Forest extracts majors, departments, courses, and prerequisites using tailored few-shot prompts with multiple LLMs. Multi-LLM cooperation mitigates information loss. Table 2 shows extraction results: 98.8% (departments), 99.5% (majors), 98.9% (courses). Deviations flagged for manual review.

#### 4.2.2 GRAPH-CONSTRUCTION AGENT

Graph-Construction Agent builds the course knowledge graph from extracted data. Construction proceeds bottom-up: department-level graphs first, then university-level integration. Cycles removed via DFS and LLM prompting. Figure 6 presents the knowledge graph. **Workflow:** (i) parse prerequisite logic, (ii) build adjacency matrix, (iii) integrate at department level, (iv) merge at university level, (v) detect/resolve cycles via DFS and LLM-based semantic analysis.

#### 4.2.3 OUTCOMES OF COURSE PLANNING AGENT

Course Planning Agent functions as a coarse planning recommender. Users specify constraints (major, completed, preferred, GE, job-related), mapped onto the knowledge graph. Figure 7 presents two example planning options. **Plan evaluation:** Generates $K = 5$ to $K = 10$ pools. Each plan

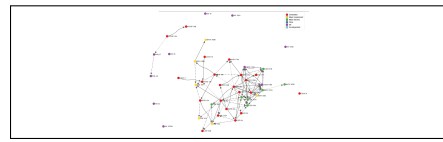
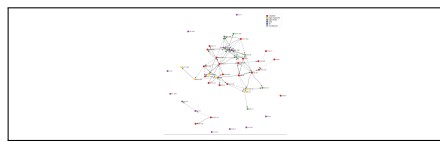

Figure 7: Coarse plan pools from Course Planning Agent. Difficulty $= \sum_c (4.0 - \text{AvgGrade}_c) \times \text{Credits}_c$. Totals: $188/136.03$ and $180/128.03$ (credits/difficulty).

| Term 1 | Term 2 | Term 3 | Term 4 | Term 5 | Term 6 |
|--------|--------|--------|--------|--------|--------|
| MATH 103A | MATH 103B | CSE 105 | CSE 158 | CSE 150A | CSE 291 |
| CSE 110 | CSE 111 | COGS 100 | COGS 118A | COGS 118B | COGS 120 |
| COGS 10 | COGS 107B | MATH 170A | MATH 180B | MATH 183 | MATH 181A |
| MATH 154 | CSE 15L | ECE 176 | COGS 1 | WCWP 10B | VIS 1 |

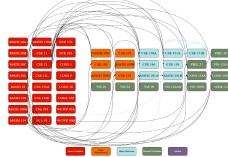

Figure 8: Baseline (left) vs. KNOWPLAN (right).

evaluated along: (i) total credits, (ii) course difficulty $\sum_c (4.0 - \text{AvgGrade}_c) \times \text{Credits}_c$ (normalized, user-steerable), (iii) requirement coverage (major and GE).

### 4.2.4 CURRICULUM ALIGNMENT AGENT

Curriculum Alignment Agent maps courses into term/term slots with four planning styles (fast-graduation, career-driven, course-driven, discovery). Over 82% of coarse plans filtered due to time conflicts. Figure 8 compares baseline and KNOWPLAN. **Scheduling refinement:** (i) Feasibility filtering: tests pools against real offerings (82% filtered). (ii) Term assignment: CP-SAT assigns courses to specific terms/sections, optimizing for selected planning mode. Output: ranked conflict-free schedules.

### 4.2.5 TEST ENVIRONMENT

**Scalability and cost.** Experiments run on $2\times$ NVIDIA H100 GPUs (80GB). We adopt a hybrid setup: Llama3.3-70B local, GPT-5 and Qwen3-32B via cloud APIs. Institution-level extraction and KG construction are embarrassingly parallel (10–15 min per institution), so 6,072 institutions require $\sim$10 hours with 100 workers. Query-level latency is dominated by CP-SAT (0.5–2s per skeleton), with concurrent Graph-RAG queries and 200–500MB per university KG. The amortized cost is $\sim$\$0.11–0.15 per plan for new institutions and $\sim$\$0.01 once institutions are indexed. The system is updated once per new catalog/term; we rerun the Agent Forest to refresh programs, courses, and policies, then rebuild the Graph-RAG index. Maintenance involves manually correcting $\sim$2–5% of records for standard HTML sites and 8–12% for JavaScript-heavy sites. Each institution's KG and constraint set is versioned by (institution, catalog year, term) and checked by a regression test suite covering structural sanity (department/major counts, course coverage), policy rules (unit caps, repeat rules), and scheduler behavior (canonical student profiles).

**Reproducibility and implementation details.** Multi-LLM panel: GPT-5 (temperature=0.1 for extraction, 0.7 for preference mapping), Llama3.3-70B (temperature=0.2, local via Hugging-Face), and Qwen3-32B (temperature=0.1, Alibaba Cloud API). Multi-LLM consensus attains 98.7% extraction accuracy with 0% hallucination (vs. 7.5–12.5% for single-LLM GPT-5). Libraries: OR-Tools (v9.8.3296), NetworkX (v3.1), FAISS (v1.7.4), OpenAI SDK (v1.12.0), HuggingFace (v4.36.0), BeautifulSoup4 (v4.12.0); Python 3.10.12. Performance: CP-SAT $\approx$1.2s per pool; Graph-RAG: 0.8ms query latency, 94.2% recall@50. Llama3.3-70B is loaded via `meta-llama/Llama-3.3-70B-Instruct`. Hyperparameters: node scoring $\lambda_1 = 0.4, \lambda_2 = 0.3, \lambda_3 = 0.3$; plan scoring $\mu_1 = 1.0, \mu_2 = 0.8, \nu_1 = 10.0, \nu_2 = 0.5, \nu_3 = 0.3$; Graph-RAG top-$K = 50$; CP-SAT timeout 30s; random seed=42.

### 4.2.6 ABLATION STUDIES AND COMPONENT EFFICIENCY

**Agent-specific efficiency analysis.** Agent Forest: $t_{\text{extract}} = 0.15 \times |\text{courses}| + 2.3$s per institution (mean 45s for 300-course catalog). Graph-Construction: 12–18 minutes for 300-course catalog.

Table 3: Comparison with SOTA

| Method | Model | KG | RAG | Data source | Heterog. | Recommendation | Dyn. | Cross |
|--------|-------|-----|-----|-------------|----------|----------------|------|-------|
| Ellucian DGW | Rule eng. | (rules) | – | Prop. DB | No | Yes (auto plans) | Yes | No |
| Stellic | Rules+heur. | (rules) | – | Prop. DB | No | Yes (next steps) | Yes | No |
| Coursicle | Heur. | – | – | Public | No | – | Yes (alerts) | Yes |
| Series25 | Opt. eng. | – | – | Prop. DB | No | – (admin sched.) | Yes | No |
| KNOWPLAN | multi-LLMs | ✓ | ✓ | Public | Yes | Coarse-to-Fine | Yes | Yes |

Features of KNOWPLAN vs. real platforms. "Prop. DB" = institution SIS/degree-audit database; "Public" = open catalogs/SOC., Abbrev.: DGW = Degree Works; Heur. = heuristics; Opt. eng. = optimization engine; Dyn. = dynamic updates; Cross = cross-institution support; Prop. = proprietary.

Course Planning: 0.8ms per Graph-RAG query, $K \times 1.2$s for top-$K$ enumeration. Curriculum Alignment: 1.2s per pool. End-to-end latency: 8–12s per student profile.

### 4.3 COMPARISON

**End-to-end system evaluation.** We evaluate KNOWPLAN on 400+ majors across four universities. Results: (i) Extraction accuracy: 95–99% precision/recall. (ii) Graph construction: 98.7% precision, 97.9% recall, 98.3% $F_1$. (iii) Planning feasibility: 94.2% of student profiles. (iv) Scheduling success: 86.1% of feasible pools (82% filtered due to time conflicts). (v) Policy compliance: 100% compliance rate. We compare KNOWPLAN with state-of-the-art methods (Table 2). Ellucian DGW Ellucian (2025a;b) and Stellic Stellic (2025) are rule-engine–driven over SIS data. Coursicle Coursicle (2025) provides schedule planning. Series25 CollegeNET (2025a;b) focuses on admin scheduling. KNOWPLAN is the only system building an explicit KG with LLM-based RAG, operating on public catalogs and addressing cross-catalog heterogeneity.

**Limitation.** Direct comparisons with Series25 limited by budget/access constraints. Comparison based on public documentation. Series25 focuses on admin scheduling (room/time optimization) vs. student-facing planning. CP-SAT ablation (Sec. 4.2.6) shows superior compliance (0 vs. 1.8 violations/plan).

## 5 CONCLUSIONS AND FUTURE WORK

**This work presents KNOWPLAN, a proactive, self-evolving multi-agent platform for adaptive and constraint-aware course planning based on publicly available university catalogs. By integrating LLM-based extraction, hypergraph construction, and term-level scheduling, KNOWPLAN bridges the gap between heterogeneous data and personalized curriculum planning. A central contribution is the design of the Agent Forest, a multi-LLM framework that addresses structural heterogeneity in course webpages. Building on this, we introduce the Logic Adjacency Matrix for modeling complex OR-AND prerequisite logic, and develop the Multi-Dependency Critical Path Extraction algorithm to support effective course sequencing in hypergraph settings. These innovations are embedded into a modular pipeline composed of four agents: Agent Forest, Graph-Construction Agent, Course Planning Agent, and Curriculum Alignment Agent. Experimental outcomes demonstrate outstanding performance over existing methods. Future work includes scaling the Agent Forest to support more universities, tracking changes in prerequisite structures over time, enhancing personalization—especially for career-aligned planning—and conducting real-world deployment to evaluate system effectiveness in practical settings.**

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
