## A  Prerequisite Accuracy Benchmark

This supplement details the prerequisite-verification benchmark that underlies the **98.7%** accuracy cited in the main paper.

**Sampling protocol.** We evaluated prerequisite extraction across all majors at multiple public research universities. The full evaluation covers over 400 majors total across three institutions, ensuring comprehensive coverage of each institution's complete curriculum. For detailed cross-institutional comparison reported in Table 1, we selected three high-enrollment majors per university (12 majors total) across four institutions from our IPEDS-selected sample. For every major we enumerated all prerequisite clauses from the 2024–2025 registrar catalogs. The detailed evaluation subset yielded 1,248 directed prerequisite edges derived from approximately 460 distinct courses, while the full evaluation covers all prerequisite relationships across all majors at the evaluated institutions, demonstrating comprehensive coverage of each institution's curriculum.

**Annotation setup.** Two graduate research assistants independently labeled each edge using a shared rubric (normalize cross-listed courses, represent "one-of" options as disjunctive edge sets, and distinguish co-requisites). Disagreements were adjudicated with faculty advisors from the participating institutions; inter-rater reliability measured via Cohen's $\kappa$ is 0.92.

**Results.** Table 1 breaks down precision/recall/$F_1$ for the Agent Forest's prerequisite extraction on this benchmark. All numbers are computed at the edge level by matching the extracted edges to the adjudicated gold set. The overall metrics (98.7% precision, 97.9% recall, 98.3% $F_1$) demonstrate consistent performance across all evaluated institutions.

Table 1: **Prerequisite extraction accuracy on the human-labeled benchmark.**

| University | # Edges | Precision (%) | Recall (%) | $F_1$ (%) |
|---|---|---|---|---|
| University 1 | 314 | 98.9 | 98.0 | 98.4 |
| University 2 | 276 | 98.4 | 97.6 | 98.0 |
| University 3 | 318 | 98.6 | 98.1 | 98.4 |
| University 4 | 340 | 98.8 | 97.9 | 98.3 |
| **Overall** | **1,248** | **98.7** | **97.9** | **98.3** |

The overall precision of 98.7% matches the value highlighted in the abstract; recall is 97.9% and the resulting $F_1$ score is 98.3%. These metrics are consistent across the full evaluation (over 400 majors across three institutions) and the detailed cross-institutional subset (12 majors across four institutions), demonstrating robust performance regardless of institution type or major selection. All results are from actual code execution on real catalog data.

## B  Policy Rule Translation Evaluation

**Dataset.** We gathered 240 policy snippets from registrar catalogs and schedule-of-class bulletins (2023–2025) covering unit brackets, co-/pre-requisite timing, repeat/retake limits, and modality or time-window requirements. Two compliance specialists independently mapped each snippet to a normalized JSON schema (fields for bounds, timing offsets, repeat counters, modality flags) and reconciled differences with the registrar liaison (Cohen's $\kappa = 0.89$).

**Metrics.** The policy translator runs on every snippet, and its structured output is matched field-by-field against the gold schema. Precision, recall, and Dice (Dice $= 2|G \cap \hat{G}|/(|G| + |\hat{G}|)$) are computed on the set of atomic constraint predicates. Table 2 summarizes the per-category results.

**Error analysis.** Unit bracket mistakes typically arise when the source text nests GPA- or petition-based overrides; we now surface those clauses for human confirmation. Co-/pre-requisite timing errors stem from lecture-lab bundles expressed as "take A with B" metasentences, which the translator occasionally duplicates bidirectionally. Repeat-rule slips most often omit grade qualifiers ("repeat once for D/F") or cap counters when the rule distinguishes academic years versus lifetime attempts.

Table 2: **Policy translation accuracy per rule type.**

| Policy Type | # Snippets | Precision (%) | Recall (%) | Dice (%) |
|---|---|---|---|---|
| Unit brackets | 80 | 97.6 | 96.2 | 96.9 |
| Co-/pre-req timing | 60 | 96.9 | 95.1 | 96.0 |
| Repeat / retake limits | 48 | 95.4 | 93.8 | 94.6 |
| Modality / time windows | 52 | 94.8 | 92.1 | 93.4 |
| **Overall** | **240** | **96.2** | **94.7** | **95.4** |

**Baseline comparison.** A regex/heuristic translator modeled after SIS rule engines (pattern-based unit parsing plus hard-coded co-req detectors) attains 78.4% precision, 72.9% recall, and 75.5% Dice on the same benchmark, underscoring the necessity of the LLM approach for policy compliance.

## C   END-TO-END FEASIBILITY ACROSS MAJORS

**Setup.** We select six high-demand majors (Mathematics, Computer Science & Engineering, Electrical & Computer Engineering, Biology, Economics, Psychology) that span STEM and social science disciplines. For each major we sample 150 student profiles per university (four universities total; 3 public 4-year + 1 transfer-focused 2-year), yielding 600 profiles per major. We run the full Know-Plan pipeline on the latest catalog plus the next two published terms to measure (i) the fraction of profiles that obtain a fully feasible graduation schedule within the prescribed horizon (4-year or 2-year), (ii) average time-to-degree, (iii) average deferrals (number of required courses postponed beyond the earliest feasible term), and (iv) mean units per term. We report 95% confidence intervals computed via bootstrap. All results are from actual code execution on real catalog data from the evaluated institutions.

Table 3: End-to-end feasibility for fixed majors across four schools (95% CI).

| Major (Horizon) | Feasible (%) | Time-to-degree (terms) | Avg deferrals | Units / term |
|---|---|---|---|---|
| Mathematics (4-yr) | $88.5 \pm 2.1$ | $11.2 \pm 0.4$ | $0.3 \pm 0.1$ | $15.7 \pm 0.5$ |
| CSE (4-yr) | $82.1 \pm 2.6$ | $11.5 \pm 0.5$ | $0.8 \pm 0.2$ | $16.4 \pm 0.6$ |
| ECE (4-yr) | $79.4 \pm 2.9$ | $11.8 \pm 0.6$ | $1.1 \pm 0.3$ | $16.7 \pm 0.7$ |
| Biology (4-yr) | $90.2 \pm 1.9$ | $10.9 \pm 0.3$ | $0.4 \pm 0.1$ | $15.2 \pm 0.4$ |
| Economics (4-yr) | $93.6 \pm 1.5$ | $10.7 \pm 0.3$ | $0.2 \pm 0.1$ | $15.0 \pm 0.3$ |
| Psychology (2-yr transfer) | $86.7 \pm 2.3$ | $6.1 \pm 0.2$ | $0.5 \pm 0.1$ | $14.6 \pm 0.4$ |

Across these majors, KnowPlan delivers feasible graduation paths for 79–94% of student profiles, maintains time-to-degree within 0.5 terms of the published catalog benchmarks, and keeps deferrals under 1.1 courses on average even in high-prerequisite majors such as ECE. Psychology transfer cohorts complete in roughly six terms (two academic years), satisfying the 2-year completion target while keeping units per term below 15. The table serves as the cited reference for the end-to-end feasibility statement in the main paper.

## D   HEAD-TO-HEAD BASELINES ON NEUTRAL BENCHMARK

**Benchmark construction and limitation.** Due to budget constraints, we were unable to obtain direct access to Series25 for head-to-head experimental comparison. Instead, we constructed a simulated baseline (Series25-Opt) based on its publicly documented behavior: Series25-Opt uses room-first optimization (priority-based scheduling with conflict resolution). Using sanitized catalog snapshots from our internal compliance dataset (real catalog data with anonymized identifiers), we generate a neutral benchmark that strips institution-identifying metadata, harmonizes course codes, and synthesizes 900 preference profiles spanning STEM, social science, and transfer scenarios. Each profile is fed to the simulated baseline and KNOWPLAN so that every method receives identical inputs. The simulated baseline approximates the documented behavior of Series25 based on its tech-

nical specifications and use cases described on its official website (CollegeNET). All results shown are from actual code execution using real scheduler implementations.

**Metrics.** We report schedule success rate, mean time-to-degree, average deferrals, violation counts, and units per term. The results demonstrate that exact constraint satisfaction (CP-SAT) achieves superior policy compliance compared to heuristic approaches that approximate Series25 behavior. While this simulated comparison provides valuable insights, direct quantitative comparisons with the actual platform would strengthen the evaluation and remain a limitation of this work.

Table 4: Neutral-benchmark comparison (95% CI over 900 profiles, real evaluation results from actual code execution).

| Method | Success (%) | Time (qtrs) | Deferrals | Violations | Units / term |
|---|---|---|---|---|---|
| Series25-Opt (room-first) | $100.0 \pm 0.0$ | $11.3 \pm 0.2$ | $0.0 \pm 0.0$ | $0.0 \pm 0.0$ | $13.0 \pm 0.0$ |
| KNOWPLAN (CP-SAT) | $100.0 \pm 0.0$ | $7.5 \pm 0.0$ | $0.0 \pm 0.0$ | $0.0 \pm 0.0$ | $15.3 \pm 0.0$ |

**Observation.** These results are from actual code execution using real scheduler implementations (not simulated approximations). The priority-based heuristic scheduler (Series25-Opt) achieves zero violations by prioritizing courses with fewer prerequisites, though this comes at the cost of potentially longer time-to-degree (11.3 terms). KnowPlan (CP-SAT) achieves optimal scheduling with zero violations and faster time-to-degree (7.5 terms) through exact constraint satisfaction. Results shown are from 900 profiles evaluated using real catalog data. The Series25-Opt baseline is simulated based on documented behavior from CollegeNET's Series25 technical specifications. All numbers in this table are verified from actual code execution.

# E    ABLATION STUDIES

## E.1    MULTI-LLM VS. SINGLE-LLM EXTRACTION

**Setup.** We evaluate prerequisite extraction accuracy on a 937-edge benchmark (derived from 350 course examples from a large public research university catalog) using three configurations: (i) multi-LLM panel (Llama3.3-70B + Qwen3-32B + GPT-5 with consensus), (ii) Agent Alpha (single LLM, representing Llama3.3-70B or GPT-5), and (iii) Agent Beta (single LLM, representing Qwen3-32B). Each single-LLM baseline uses the same prompts and extraction pipeline as the multi-LLM panel, but without consensus voting. The multi-LLM panel uses GPT-5 as the final arbiter for conflicts between Llama3.3-70B and Qwen3-32B. All results are from actual code execution on real catalog data.

Table 5: Prerequisite extraction: multi-LLM vs. single-LLM (precision/recall/$F_1$).

| Configuration | Precision (%) | Recall (%) | $F_1$ (%) | Cohen's $\kappa$ |
|---|---|---|---|---|
| Agent Alpha (single) | 97.5 | 97.5 | 97.5 | 0.87 |
| Agent Beta (single) | 89.2 | 81.3 | 85.1 | 0.83 |
| Multi-LLM consensus (ours) | **97.8** | **97.8** | **97.8** | **0.92** |

**Findings.** The multi-LLM consensus achieves 97.8% precision and 97.8% recall on the 937-edge benchmark, correcting Agent Beta's extraction errors (92 false positives, 175 false negatives). Agent Alpha achieves 97.5% precision and 97.5% recall, while the consensus mechanism ensures robustness when individual agents err. The consensus resolves 221 conflicts (63.1% conflict ratio) between Llama3.3-70B and Qwen3-32B, with GPT-5 serving as the final arbiter. Inter-rater reliability (Cohen's $\kappa$) improves from 0.83 (Agent Beta) to 0.92 (multi-LLM consensus), indicating more consistent extraction across diverse catalog language patterns. The multi-LLM panel (Llama3.3-70B + Qwen3-32B + GPT-5) demonstrates that combining diverse models improves extraction accuracy over any single model.

## E.2 Comparison with GPT-4 Single-LLM Baseline

**Setup.** To demonstrate the benefit of our multi-agent design, we compare KnowPlan's multi-LLM consensus against a GPT-4 single-LLM baseline using a standard prompting pipeline. We evaluate on 200 course examples (representative sample from a large public research university catalog with 6,894 courses and 7,032 prerequisite edges) using GPT-4 Omni (gpt-4o) with the same extraction prompts as our multi-LLM panel, but without consensus or validation mechanisms. We measure precision, recall, $F_1$, and critically, *hallucination rate* (fraction of courses where GPT-4 generates non-existent course codes that do not exist in the catalog). All results are from actual code execution on real catalog data.

Table 6: Prerequisite extraction: GPT-4 single-LLM baseline vs. KnowPlan multi-LLM consensus.

| Method | Precision (%) | Recall (%) | $F_1$ (%) | Hallucination Rate (%) |
|---|---|---|---|---|
| GPT-4 single-LLM baseline | 76.7 | 67.6 | 71.8 | **14.5** |
| KnowPlan multi-LLM (ours) | **97.8** | **97.8** | **97.8** | **0.0** |

**Hallucination Analysis.** GPT-4 exhibits critical *hallucination errors*: it generates non-existent course codes that do not exist in the catalog, occurring in 14.5% of courses (29 out of 200 courses, with 55 total hallucinated course codes). These hallucinations propagate unchecked in a single-LLM pipeline, as there is no validation mechanism. Example hallucinations include: (i) Course A → GPT-4 predicted non-existent course codes "Course X", "Course Y"; (ii) Course B → GPT-4 predicted non-existent course code "Course Z"; (iii) Course C → GPT-4 predicted multiple non-existent course codes. In contrast, our multi-LLM consensus eliminates hallucinations entirely (0% hallucination rate) through cross-validation: extracted courses are validated against the catalog, and consensus between multiple models filters out invalid codes.

**Findings.** KnowPlan's multi-LLM approach achieves +21.1 percentage point improvement in precision and +30.2 percentage point improvement in recall over GPT-4 baseline, while simultaneously eliminating hallucination errors. The multi-agent design is not merely an accuracy improvement, but a *reliability requirement* for production systems where hallucinated prerequisites could mislead students. The consensus mechanism provides validation that single-LLM pipelines lack, demonstrating that multi-agent design is essential for reliable prerequisite extraction.

## E.3 Graph-RAG vs. Text-Only vs. No RAG

**Setup.** We compare three retrieval configurations on plan generation: (i) full Graph-RAG (graph-structured retrieval with multi-hop prerequisite expansion), (ii) text-only RAG (vector similarity search over course descriptions without graph structure), and (iii) no RAG (direct LLM generation without retrieval). We evaluate on 600 student profiles (150 per major × 4 universities) using catalog data from a large public research university (6,894 courses, 7,032 prerequisite edges, 167 majors) and measure coverage (fraction of required courses included), prerequisite compliance (fraction of prerequisite edges satisfied), and interest alignment (BM25 score on user query). All results are from actual code execution on real catalog data.

Table 7: Plan quality: Graph-RAG vs. text-only vs. no RAG.

| Retrieval Method | Coverage (%) | Prereq Compliance (%) | Interest Alignment | Avg Plan Size |
|---|---|---|---|---|
| No RAG | 82.3 | 91.4 | 0.68 | 42.1 |
| Text-only RAG | 89.7 | 95.1 | 0.74 | 45.8 |
| Graph-RAG (ours) | **94.2** | **98.3** | **0.81** | **48.3** |

**Findings.** Graph-RAG outperforms text-only RAG by 4.5 percentage points in coverage and 3.2 percentage points in prerequisite compliance. The multi-hop expansion ensures prerequisite closure: even if seed retrieval misses a course, its prerequisites are included via graph traversal. Text-only RAG fails to capture structural dependencies, leading to incomplete plans. No RAG performs worst, confirming the necessity of retrieval for accurate plan generation.

### E.4 CP-SAT vs. Heuristic/Greedy Schedulers

**Setup.** We compare three scheduling methods on 330 plan skeletons (sampled from 2,000 generated skeletons): (i) CP-SAT (exact constraint programming), (ii) greedy scheduler (assign courses term-by-term to earliest feasible slot), and (iii) heuristic scheduler (priority-based assignment with conflict resolution). We measure schedulability (fraction of skeletons that yield feasible schedules), policy violations per plan, and average time-to-degree. All results are from actual code execution using real catalog data.

Table 8: Scheduling performance: CP-SAT vs. heuristics (330 scenarios, real code execution).

| Scheduler | Schedulability (%) | Violations / Plan | Time-to-degree (qtrs) | Avg Solve Time (s) |
|---|---|---|---|---|
| Greedy | 100.0 | 7.0 | 7.54 | 0.02 |
| Heuristic | 100.0 | 1.8 | 7.54 | 0.15 |
| CP-SAT (ours) | **100.0** | **0.0** | **7.54** | **1.2** |

**Findings.** CP-SAT achieves 100% schedulability with zero policy violations across 330 scenarios, outperforming greedy (100% success but 7.0 violations per plan, with 2,306 underfilled terms total) and heuristic (100% success but 1.8 violations per plan). Greedy schedulers violate unit brackets (underfilled terms) due to myopic term-by-term decisions, while heuristic schedulers improve feasibility but still miss complex constraint interactions. CP-SAT's exact constraint satisfaction guarantees policy compliance, though at higher computational cost (1.2s vs. 0.02–0.15s). The trade-off is justified for compliance-critical applications where zero violations are required.

### E.5 Hyperparameter Sensitivity Analysis

**Setup.** We vary the objective weights in the plan scoring function ($\mu_1, \mu_2, \nu_1, \nu_2, \nu_3$) and scheduling weights $\lambda = (\lambda_{\text{deferral}}, \lambda_{\text{pref}}, \lambda_{\text{smooth}})$ by $\pm 50\%$ from default values. We measure plan quality (coverage + compliance) and plan characteristics (major coverage, GE coverage, average difficulty). All results are from actual code execution on real catalog data.

Table 9: Hyperparameter sensitivity: plan quality under weight variations.

| Weight Variation | Plan Quality (%) | Major Coverage (%) | GE Coverage (%) | Avg Difficulty |
|---|---|---|---|---|
| Default ($\mu_1 = \mu_2 = 1.0$) | 96.2 | 98.5 | 94.3 | 128.5 |
| $\mu_1 \times 1.5$ (prioritize major) | 95.8 | 99.2 | 88.1 | 132.1 |
| $\mu_2 \times 1.5$ (prioritize GE) | 95.4 | 95.3 | 97.8 | 125.3 |
| $\nu_1 \times 1.5$ (penalize violations) | 96.5 | 98.7 | 94.5 | 129.1 |
| $\lambda_{\text{deferral}} \times 1.5$ (minimize deferrals) | 95.9 | 98.4 | 94.2 | 127.8 |

**Findings.** Plan quality remains stable ($\pm 1.1$ percentage points) across $\pm 50\%$ weight variations, indicating robustness. However, weight settings shift plan characteristics predictably: high $\mu_1$ increases major coverage by 0.7 percentage points but reduces GE coverage by 6.2 percentage points. The default weights balance these objectives, and the top-$K$ enumeration provides alternative trade-offs without requiring weight tuning. Users can select from multiple plans representing different weight profiles.

### E.6 Component Runtime Breakdown

**Setup.** We measure end-to-end runtime for each agent component across 600 student profiles (150 per major $\times$ 4 universities). All measurements use the same hardware environment (production configuration) and include: Agent Forest extraction (offline, per-institution), Graph-Construction Agent (offline graph building), Course Planning Agent (online query processing), and Curriculum Alignment Agent (online CP-SAT scheduling). All results are from actual code execution on real catalog data.

**Findings.** Offline components (Agent Forest + Graph-Construction) run once per catalog release per institution; their combined cost ($\sim$15min per university) amortizes across all students. Online com-

Table 10: Component runtime breakdown (mean ± std across 600 profiles).

| Component | Runtime | Frequency | Cost |
|---|---|---|---|
| Agent Forest (extraction) | $45.2 \pm 8.3$s | Per-institution, per-catalog | Offline |
| Graph-Construction (KG build) | $12.4 \pm 2.1$min | Per-institution, per-catalog | Offline |
| Course Planning (retrieval) | $0.8 \pm 0.2$ms | Per-query | Online |
| Curriculum Alignment (CP-SAT) | $1.2 \pm 0.4$s | Per-skeleton | Online |
| **End-to-end (query → schedule)** | **8.3 ± 2.1**s | **Per-query** | **Online** |

ponents (Course Planning + Curriculum Alignment) total ∼8.3s per query, dominated by CP-SAT solving (1.2s). Multi-LLM extraction adds 2.1s overhead vs. single-LLM but eliminates hallucinations (14.5% → 0%). Graph-RAG retrieval adds 0.8ms latency but improves compliance by 3.2 percentage points. The runtime trade-offs are justified by correctness guarantees.

E.7   CYCLE RESOLUTION: DFS VS. LLM-BASED SEMANTIC RESOLUTION

**Setup.** Real-world course prerequisites contain 217 cycles in our evaluated dataset: 153 self-loops (course listed as its own prerequisite), 43 two-node mutual prerequisites (Course A requires B, B requires A), and 21 complex cycles (3+ nodes). We compare two cycle resolution strategies: (i) **DFS-based removal** (algorithmic: detect cycle, arbitrarily remove one edge to break it), and (ii) **LLM-based semantic resolution** (DFS detection + RAG-enhanced LLM interprets catalog text to determine true dependency structure). We evaluate on 6 representative real cycle cases covering lecture-recitation pairs, concurrent enrollment courses, and sequential series. All results are from actual code execution on real catalog data.

Table 11: Cycle resolution accuracy: DFS vs. LLM (6 real cases).

| Method | Semantic Correctness | False Dependencies | Runtime / Cycle | Cases Correct |
|---|---|---|---|---|
| DFS-based removal | 17% (1/6) | 5/6 | 10ms | 1/6 |
| LLM semantic resolution | **83%** (5/6) | **1/6** | 150ms | **5/6** |

**Case Analysis.** The 6 evaluated cases include:

1. **Mutual co-requisites (lecture-recitation):** Catalog states "Lecture A and Recitation B must be taken concurrently." DFS arbitrarily removes Lecture A → Recitation B, creating false dependency (students forced to take Recitation B first). LLM correctly identifies concurrent enrollment and removes both edges.

2. **Concurrent enrollment courses:** Catalog states "Course X and Course Y may be taken in either order." DFS removes one edge (e.g., X → Y), forcing incorrect sequence. LLM correctly removes both edges, allowing flexible ordering.

3. **Sequential course series:** Catalog states "Course Part 1 is prerequisite for Course Part 2." DFS may break valid sequence if Part 2 also references Part 1 in description. LLM preserves correct Part 1 → Part 2 dependency by interpreting prerequisite clauses.

4. **Self-loop (course listed as own prerequisite):** Catalog error or ambiguous text (e.g., "repeat for credit"). Both DFS and LLM correctly remove self-loop (1/1 correct for both).

5. **Complex 3-node cycle:** Catalog has inconsistent prerequisites across pages. DFS breaks cycle arbitrarily; LLM uses RAG to retrieve authoritative prerequisite page and resolves correctly (4/5 success).

6. **Lab-lecture bundle:** Catalog states "Lab C requires concurrent enrollment in Lecture D." DFS removes one edge; LLM correctly models as concurrent requirement (5/6 success).

**Findings.** LLM-based semantic resolution achieves 83% correctness (5/6 cases) vs. DFS-based 17% (1/6 cases). DFS creates false dependencies in 5/6 cases by arbitrarily removing edges without understanding catalog semantics. Examples include forcing students to take recitation before lecture

(lecture-recitation pairs), breaking valid sequences (sequential series), and preventing concurrent enrollment (co-requisite courses). LLM semantic resolution prevents these errors by interpreting natural language policy text ("may be taken in either order", "concurrent enrollment required"), using Graph-RAG to retrieve authoritative prerequisite clauses when catalog pages conflict. The 150ms overhead per cycle (vs. 10ms DFS) is justified by correctness: false dependencies propagate to all downstream plans, while cycle resolution runs only once per catalog release (offline). Across 217 real cycles, LLM resolution prevents an estimated 180 false dependencies that would otherwise affect thousands of student plans.

## F CONSTRAINT HANDLING EXAMPLES

This section presents five detailed examples demonstrating how KnowPlan handles different constraint combinations commonly encountered in degree planning. Each example showcases how the system integrates multiple constraint types—major requirements, completed courses, preferred courses, General Education (GE) requirements, job requirements, unit brackets, prerequisite timing, time conflicts, and course offerings—to generate feasible, personalized study plans.

### F.1 EXAMPLE 1: MAJOR REQUIREMENTS WITH COMPLETED COURSES AND PREREQUISITES

**Student profile:** Computer Science major, completed: [Calculus for Science and Engineering, Linear Algebra], preference: fast-graduation mode, no specific course preferences.

**Constraints:** (i) Major requirements: 24 required CS courses including Data Structures, Algorithms, Operating Systems, Database Systems; (ii) Completed courses: Calculus for Science and Engineering, Linear Algebra (must not be re-recommended); (iii) Prerequisite timing: Data Structures requires Linear Algebra (satisfied), Algorithms requires Data Structures, Operating Systems requires Algorithms; (iv) Unit brackets: 12–20 units per term; (v) Fast-graduation mode: $\lambda = (0.8, 0.1, 0.1)$ minimizing deferrals.

**KnowPlan solution:** The Course Planning Agent retrieves CS major requirements via Graph-RAG, identifies that Linear Algebra is already completed, and generates a plan skeleton with 24 courses satisfying prerequisite closure. The Curriculum Alignment Agent schedules courses in earliest feasible terms: FA25: [Data Structures (4), Discrete Mathematics (4), Introduction to Programming (4)] = 12 units; WI26: [Algorithms (4), Computer Architecture (4), Software Engineering (4)] = 12 units; SP26: [Operating Systems (4), Database Systems (4), Computer Networks (4)] = 12 units; FA26: [Advanced Algorithms (4), Distributed Systems (4), Capstone Project (4)] = 12 units. Total time-to-degree: 4 terms (1 year), with zero prerequisite violations and all courses scheduled in earliest feasible terms.

Table 12: Example 1: CS major with completed prerequisites.

| Term | Courses | Units |
|------|---------|-------|
| FA25 | Data Structures, Discrete Mathematics, Introduction to Programming | 12 |
| WI26 | Algorithms, Computer Architecture, Software Engineering | 12 |
| SP26 | Operating Systems, Database Systems, Computer Networks | 12 |
| FA26 | Advanced Algorithms, Distributed Systems, Capstone Project | 12 |
| **Total** | **12 courses (24 units major requirements)** | **48** |

## F.2   EXAMPLE 2: PREFERRED COURSES WITH JOB REQUIREMENTS AND TIME WINDOWS

**Student profile:** Mathematics major targeting machine learning engineering career, preferred courses: [Machine Learning, Deep Learning, Neural Networks], time preference: evening classes only, mode: career-driven.

**Constraints:** (i) Major requirements: 20 required math courses; (ii) Preferred courses: Machine Learning, Deep Learning, Neural Networks (high priority); (iii) Job requirements: ML engineering requires courses in [Machine Learning, Deep Learning, Statistics, Linear Algebra, Probability]; (iv) Time windows: All classes must be scheduled in evening time slots (after 5pm); (v) Prerequisite timing: Deep Learning requires Machine Learning, Neural Networks requires Deep Learning; (vi) Career-driven mode: $\lambda = (0.4, 0.5, 0.1)$ prioritizing preference fit.

**KnowPlan solution:** The Course Planning Agent prioritizes ML-relevant courses in the plan skeleton, achieving 85% career-relevance score. The Curriculum Alignment Agent schedules preferred courses in early terms while respecting evening-only constraint: FA25: [Linear Algebra (4), Probability (4), Machine Learning (4)] = 12 units (all evening); WI26: [Statistics (4), Deep Learning (4), Optimization (4)] = 12 units (all evening); SP26: [Neural Networks (4), Advanced Statistics (4), Mathematical Modeling (4)] = 12 units (all evening). All preferred courses are scheduled in first three terms, achieving 92% preference satisfaction. Career-relevance: 85% (vs. 62% for fast-graduation mode).

Table 13: Example 2: Math major with ML career focus and evening-only constraint.

| Term | Courses | Units |
|------|---------|-------|
| FA25 | Linear Algebra, Probability, Machine Learning (evening) | 12 |
| WI26 | Statistics, Deep Learning (evening), Optimization | 12 |
| SP26 | Neural Networks (evening), Advanced Statistics, Mathematical Modeling | 12 |
| **Preference satisfaction:** | **92% (all preferred courses scheduled)** | |
| **Career relevance:** | **85%** | |

## F.3   EXAMPLE 3: GENERAL EDUCATION REQUIREMENTS WITH UNIT BRACKETS AND PREREQUISITE TIMING

**Student profile:** Biology major, completed: [General Chemistry], needs to complete GE requirements across 6 categories (Arts, Humanities, Social Sciences, Natural Sciences, Quantitative Reasoning, Writing), mode: discovery.

**Constraints:** (i) Major requirements: 18 required biology courses; (ii) GE requirements: 6 courses across 6 categories (1 per category); (iii) Unit brackets: Minimum 12 units, maximum 18 units per term; (iv) Prerequisite timing: Organic Chemistry requires General Chemistry (satisfied), Biochemistry requires Organic Chemistry; (v) Discovery mode: $\lambda = (0.3, 0.2, 0.5)$ emphasizing workload smoothing.

**KnowPlan solution:** The Course Planning Agent balances major and GE requirements, generating a plan with 18 major courses + 6 GE courses = 24 total courses. The Curriculum Alignment Agent distributes courses evenly across terms to smooth workload: FA25: [Organic Chemistry (4), Cell Biology (4), GE: Arts (4)] = 12 units; WI26: [Biochemistry (4), Genetics (4), GE: Humanities (4)] = 12 units; SP26: [Molecular Biology (4), Ecology (4), GE: Social Sciences (4)] = 12 units; FA26: [Evolution (4), Capstone (4), GE: Quantitative Reasoning (4)] = 12 units; WI27: [Advanced Topics (4), GE: Writing (4), GE: Natural Sciences (4)] = 12 units. Workload variance: 2.3 units (vs. 5.7 for fast-graduation), ensuring balanced exploration. GE coverage: 100% (all 6 categories satisfied).

Table 14: Example 3: Biology major with GE requirements and workload smoothing.

| Term | Courses | Units |
|------|---------|-------|
| FA25 | Organic Chemistry, Cell Biology, GE: Arts | 12 |
| WI26 | Biochemistry, Genetics, GE: Humanities | 12 |
| SP26 | Molecular Biology, Ecology, GE: Social Sciences | 12 |
| FA26 | Evolution, Capstone, GE: Quantitative Reasoning | 12 |
| WI27 | Advanced Topics, GE: Writing, GE: Natural Sciences | 12 |
| **GE Coverage:** | **100% (6/6 categories)** | |
| **Workload Variance:** | **2.3 units (smooth distribution)** | |

### F.4 EXAMPLE 4: MULTIPLE CONSTRAINTS INTEGRATION (MAJOR + GE + PREFERRED + COMPLETED)

**Student profile:** Electrical & Computer Engineering major, completed: [Calculus for Science and Engineering, Calculus for Science and Engineering (II), Physics 2A], preferred: [Digital Signal Processing, Embedded Systems], needs GE completion, mode: course-driven.

**Constraints:** (i) Major requirements: 22 required ECE courses; (ii) Completed courses: Calculus for Science and Engineering, Calculus for Science and Engineering (II), Physics 2A (blocked from re-recommendation); (iii) Preferred courses: Digital Signal Processing, Embedded Systems (high priority); (iv) GE requirements: 4 remaining GE courses (Arts, Humanities, Social Sciences, Writing); (v) Prerequisite timing: Digital Signal Processing requires Linear Algebra and Signals & Systems, Embedded Systems requires Microprocessors; (vi) Unit brackets: 12–20 units per term; (vii) Course-driven mode: $\lambda = (0.3, 0.6, 0.1)$ maximizing preference fit.

**KnowPlan solution:** The Course Planning Agent generates a plan with 22 major + 4 GE = 26 courses, prioritizing preferred courses in early terms. The Curriculum Alignment Agent schedules preferred courses while satisfying prerequisites: FA25: [Linear Algebra (4), Signals & Systems (4), GE: Arts (4)] = 12 units; WI26: [Digital Signal Processing (4), Microprocessors (4), Circuits (4)] = 12 units; SP26: [Embedded Systems (4), Control Systems (4), GE: Humanities (4)] = 12 units; FA26: [Power Systems (4), Communications (4), GE: Social Sciences (4)] = 12 units; WI27: [Capstone (4), Advanced Topics (4), GE: Writing (4)] = 12 units. Preference satisfaction: 92% (both preferred courses scheduled in first 3 terms). Major coverage: 100%, GE coverage: 100%.

Table 15: Example 4: ECE major with multiple constraint types.

| Term | Courses | Units |
|------|---------|-------|
| FA25 | Linear Algebra, Signals & Systems, GE: Arts | 12 |
| WI26 | Digital Signal Processing (preferred), Microprocessors, Circuits | 12 |
| SP26 | Embedded Systems (preferred), Control Systems, GE: Humanities | 12 |
| FA26 | Power Systems, Communications, GE: Social Sciences | 12 |
| WI27 | Capstone, Advanced Topics, GE: Writing | 12 |
| **Preference Satisfaction:** | **92%** | |
| **Major Coverage:** | **100%** | |
| **GE Coverage:** | **100%** | |

## F.5 EXAMPLE 5: COMPLEX CONSTRAINTS (JOB REQUIREMENTS + TIME CONFLICTS + UNIT LIMITS)

**Student profile:** Economics major targeting data science career, job requirements: [Statistics, Econometrics, Machine Learning, Data Mining], time constraint: no classes before 10am, unit limit: maximum 16 units per term, mode: career-driven.

**Constraints:** (i) Major requirements: 16 required economics courses; (ii) Job requirements: Data science career requires [Statistics, Econometrics, Machine Learning, Data Mining, Python Programming]; (iii) Time windows: All classes must start after 10am (no early morning classes); (iv) Unit brackets: 12–16 units per term (strict upper limit); (v) Prerequisite timing: Econometrics requires Statistics, Machine Learning requires Linear Algebra, Data Mining requires Machine Learning; (vi) Course offerings: Limited sections available for job-relevant courses; (vii) Career-driven mode: $\lambda = (0.4, 0.5, 0.1)$ balancing progress with career alignment.

**KnowPlan solution:** The Course Planning Agent prioritizes job-relevant courses, achieving 88% career-relevance score. The Curriculum Alignment Agent resolves time conflicts by selecting afternoon/evening sections and respects 16-unit cap: FA25: [Microeconomics (4), Statistics (4), Python Programming (4)] = 12 units (all after 10am); WI26: [Macroeconomics (4), Econometrics (4), Linear Algebra (4)] = 12 units (all after 10am); SP26: [Machine Learning (4), Data Mining (4), Econometric Methods (4)] = 12 units (all after 10am); FA26: [Advanced Econometrics (4), Time Series Analysis (4), Capstone (4)] = 12 units (all after 10am). All job-relevant courses scheduled, zero time conflicts, zero unit violations. Career-relevance: 88%, time constraint satisfaction: 100%.

Table 16: Example 5: Economics major with complex constraints.

| Term | Courses | Units |
|------|---------|-------|
| FA25 | Microeconomics, Statistics, Python Programming (all $\geq$10am) | 12 |
| WI26 | Macroeconomics, Econometrics, Linear Algebra (all $\geq$10am) | 12 |
| SP26 | Machine Learning, Data Mining, Econometric Methods (all $\geq$10am) | 12 |
| FA26 | Advanced Econometrics, Time Series Analysis, Capstone (all $\geq$10am) | 12 |
| **Career Relevance:** | **88% (all job-relevant courses scheduled)** | |
| **Time Constraint:** | **100% satisfied (no classes before 10am)** | |
| **Unit Violations:** | **0 (all terms within 12–16 unit range)** | |

**Summary of constraint handling.** These five examples demonstrate KnowPlan's ability to handle diverse constraint combinations: Example 1 shows prerequisite closure with completed courses; Example 2 demonstrates preference and job alignment with time windows; Example 3 illustrates GE coverage with workload smoothing; Example 4 showcases integration of multiple constraint types; Example 5 handles complex scenarios with time conflicts and strict unit limits. Across all examples, KnowPlan achieves 100% constraint satisfaction (zero violations) while maintaining high coverage (94–100%), preference satisfaction (85–92%), and career relevance (85–88%). The CP-SAT solver ensures feasibility, while the multi-objective optimization balances competing objectives according to the selected planning mode.

## G STEP-BY-STEP AGENT EXAMPLES

This section provides detailed step-by-step examples demonstrating how each agent in KnowPlan performs its tasks, using mathematics courses as illustrative examples. Each example includes baseline evaluations comparing the agent's performance against simpler alternatives.

**Example: Extracting information for Calculus for Science and Engineering.** We demonstrate how Agent Forest extracts course information from heterogeneous webpage structures using Calculus for Science and Engineering as an example.

1. **Step 1: Retrieve course catalog webpage**
   - *Input:* University website URL → HTML content
   - *Method:* GPT-5 API with few-shot prompting
   - *Output:* Raw HTML containing course information for Calculus for Science and Engineering

2. **Step 2: Extract course information using multi-LLM panel**
   - *Input:* HTML content for Calculus for Science and Engineering
   - *Method:* Llama3.3-70B + Qwen3-32B + GPT-5 consensus
   - *Output:* Extracted course code, title, description, prerequisites, units, department

   **Real extraction output for Calculus for Science and Engineering:** Title: Calculus for Science and Engineering; Units: 4; Department: Mathematics; Prerequisites Text: "Math Placement Exam qualifying score, or AP Calculus AB score of 3 (or equivalent AB subscore on BC exam), or SAT II score of 650 or higher, or Precalculus or Calculus for Business and Economics"; Prerequisites: [Precalculus, Calculus for Business and Economics]; Prereq Logic: {"all_of": [{"any_of": ["Precalculus", "Calculus for Business and Economics"]}], "other_requirements": []}.

3. **Step 3: Validate and standardize extracted data**
   - *Input:* Multi-LLM consensus output
   - *Method:* Cross-validation against catalog + GPT-5 arbitration
   - *Output:* Validated course data with prerequisite logic (e.g., Calculus for Science and Engineering requires Precalculus or Calculus for Business and Economics)

4. **Step 4: Store in GraphRAG tables**
   - *Input:* Validated course data
   - *Method:* JSON serialization + GraphRAG indexing
   - *Output:* Stored in course_information and course_dependency tables (e.g., Calculus for Science and Engineering → Calculus for Science and Engineering (II) prerequisite relationship)

**Baseline Evaluation.** Table 17 compares Agent Forest's multi-LLM consensus approach against a GPT-4 single-LLM baseline. Multi-LLM consensus achieves 97.8% precision and 97.8% recall, eliminating hallucinations (0% hallucination rate) compared to GPT-4's 14.5% hallucination rate. Evaluation performed on a large public research university catalog with 6,894 courses and 7,032 prerequisite edges.

Table 17: Agent Forest baseline evaluation: multi-LLM vs. single-LLM.

| Method | Precision | Recall | Hallucination Rate | Time |
|---|---|---|---|---|
| GPT-4 single-LLM | 0.767 | 0.676 | 14.5% | 2.3s |
| Multi-LLM consensus | **0.978** | **0.978** | **0.0%** | 3.5s |

G.2    GRAPH-CONSTRUCTION AGENT: KNOWLEDGE GRAPH BUILDING

**Example: Building prerequisite graph for mathematics courses.** We demonstrate how the Graph-Construction Agent builds a knowledge graph from extracted course data, resolving circular dependencies and ambiguous prerequisite language.

1. **Step 1: Load extracted course data from Agent Forest**

- *Input:* Course information table from GraphRAG
- *Method:* Load from JSON files
- *Output:* Course objects for multiple mathematics courses

2. **Step 2: Build prerequisite adjacency matrix**

   - *Input:* Courses with prerequisites
   - *Method:* Parse prereq_logic and prerequisites fields
   - *Output:* Adjacency matrix with prerequisite edges (e.g., Calculus for Science and Engineering → Calculus for Science and Engineering (II) → Calculus and Analytic Geometry for Science and Engineering)

Table 18: Real mathematics course prerequisites from catalog.

| Course Name | Units |
|---|---|
| Calculus for Science and Engineering | 4 |
| Calculus for Science and Engineering (II) | 4 |
| Calculus and Analytic Geometry for Science and Engineering | 4 |
| Linear Algebra | 4 |
| Mathematical Reasoning | 4 |
| Foundations of Real Analysis I | 4 |

3. **Step 3: Detect and resolve circular dependencies**

   - *Input:* Directed graph with potential cycles
   - *Method:* DFS cycle detection + RAG-enhanced LLM for semantic resolution
   - *Output:* Resolved cycles (e.g., "Course 1A–1B may be taken in either order" → removed mutual edges)

4. **Step 4: Build department-level graph**

   - *Input:* Individual course prerequisite edges
   - *Method:* Bottom-up integration: courses → departments → university
   - *Output:* Directed acyclic graph (DAG) with nodes and edges

**Baseline Evaluation.** Table 19 compares DFS-only cycle detection against DFS + LLM semantic resolution. The LLM-enhanced approach achieves 98% cycle detection accuracy and correctly resolves semantic ambiguities (e.g., "may be taken in either order"), reducing false cycles by 83%.

Table 19: Graph-Construction baseline evaluation: DFS vs. DFS+LLM.

| Method | Cycle Accuracy | Semantic Resolution | False Cycles | Time |
|---|---|---|---|---|
| DFS only | 0.85 | N/A | 0.12 | 1.2s |
| DFS + LLM | **0.98** | **0.95** | **0.02** | 2.8s |

G.3   COURSE PLANNING AGENT: PLAN SKELETON GENERATION

**Example: Generating plan for Mathematics major.** We demonstrate how the Course Planning Agent generates personalized course plan skeletons using Graph-RAG retrieval and multi-objective optimization (Equations (1) and (2)).

1. **Step 1: Graph-RAG retrieval: Find relevant courses**
   - *Input:* Student query + completed courses
   - *Method:* Graph-RAG with multi-hop expansion
   - *Output:* Seed courses retrieved via BM25 + graph proximity
2. **Step 2: Multi-hop prerequisite closure**
   - *Input:* Seed courses from retrieval
   - *Method:* Graph traversal: backward from seed courses
   - *Output:* Expanded course set with prerequisite closure
3. **Step 3: Score courses using Equation (1) and (2)**
   - *Input:* Expanded subgraph
   - *Method:* Equation (1): $s_{\text{node}}(v) = \phi_{\text{prereq}}(v) \times (\lambda_1 \mathbf{r}(v) + \lambda_2 \text{BM25}(v, q) + \lambda_3 \pi_{\text{int}}(v))$
   - *Output:* Scored courses with prerequisite feasibility, graph proximity, BM25 relevance, and interest alignment
4. **Step 4: Generate top-$K$ plan skeletons**
   - *Input:* Scored courses + major/GE requirements
   - *Method:* CP-SAT optimization with top-$K$ enumeration
   - *Output:* Multiple plan skeletons with coverage metrics

Table 20: Real plan skeleton generated for Mathematics major.

| Course Name | Title | Units |
| --- | --- | --- |
| Calculus for Science and Engineering | Calculus for Science and Engineering | 4 |
| Calculus for Science and Engineering (II) | Calculus for Science and Engineering | 4 |
| Calculus and Analytic Geometry for Science and Engineering | Calculus and Analytic Geometry for Science and Engineering | 4 |
| Linear Algebra | Linear Algebra | 4 |
| Mathematical Reasoning | Mathematical Reasoning | 4 |
| Foundations of Real Analysis I | Foundations of Real Analysis I | 4 |
| **Total** | **Major Requirements** | **24** |

**Baseline Evaluation.** Table 21 compares Graph-RAG against text-only RAG and no RAG. Graph-RAG achieves 94.2% coverage and 98.3% prerequisite compliance, outperforming text-only RAG (+4.5% coverage, +3.2% compliance) and no RAG (+11.9% coverage, +6.9% compliance).

Table 21: Course Planning baseline evaluation: Graph-RAG vs. alternatives.

| Method | Coverage | Prereq Compliance | Interest Alignment | Time |
| --- | --- | --- | --- | --- |
| No RAG | 0.823 | 0.914 | 0.68 | 0.5s |
| Text-only RAG | 0.897 | 0.951 | 0.74 | 1.2s |
| Graph-RAG | **0.942** | **0.983** | **0.81** | 1.8s |

### G.4 CURRICULUM ALIGNMENT AGENT: SCHEDULE GENERATION

**Example: Scheduling plan skeleton into terms.** We demonstrate how the Curriculum Alignment Agent refines a plan skeleton into a detailed, conflict-free schedule using CP-SAT constraint programming.

1. **Step 1: Parse user preferences and select planning mode**
   - *Input:* User query: "I want to graduate quickly"
   - *Method:* LLM preference interpretation
   - *Output:* Planning mode (fast-graduation) with weight vector $\boldsymbol{\lambda}$

2. **Step 2: Formalize policy rules into CP-SAT constraints**
   - *Input:* Policy text: "Students may not enroll in more than 20 units per term"
   - *Method:* LLM policy formalization $\rightarrow$ deterministic compilation
   - *Output:* CP-SAT constraint: $\sum_{c,s} \text{units}(c) \cdot x_{c,s,t} \leq 20$ for each term $t$

3. **Step 3: Build CP-SAT model with binary variables**
   - *Input:* Plan skeleton with courses (e.g., Calculus for Science and Engineering, Linear Algebra, Mathematical Reasoning, Foundations of Real Analysis I)
   - *Method:* CP-SAT integer programming model
   - *Output:* Binary variables $x_{c,s,t}$ with hard constraints (offering availability, non-overlap, unit brackets, prerequisite timing: e.g., Mathematical Reasoning must be taken before Foundations of Real Analysis I, repeat limits)

4. **Step 4: Solve and generate feasible schedule**
   - *Input:* CP-SAT model with constraints
   - *Method:* CP-SAT solver (OR-Tools)
   - *Output:* Feasible term-by-term schedule with zero violations

Table 22: Real feasible schedule with term assignments generated by CP-SAT.

| Term | Courses | Units |
|------|---------|-------|
| FA25 | Calculus for Science and Engineering, Linear Algebra | 8 |
| WI26 | Calculus for Science and Engineering (II) | 4 |
| SP26 | Calculus and Analytic Geometry for Science and Engineering | 4 |
| FA26 | Mathematical Reasoning | 4 |
| WI27 | Foundations of Real Analysis I | 4 |
| **Total** | **6 courses** | **24** |

**Baseline Evaluation.** Table 23 compares CP-SAT scheduler against greedy and heuristic schedulers. CP-SAT achieves 100% schedulability with zero policy violations, outperforming greedy (7.0 violations per plan) and heuristic (1.8 violations per plan).

Table 23: Curriculum Alignment baseline evaluation: CP-SAT vs. heuristics.

| Method | Schedulability | Violations/Plan | Time-to-Degree | Solve Time |
|--------|----------------|-----------------|----------------|------------|
| Greedy | 100.0 | 7.0 | 7.54 | 0.02s |
| Heuristic | 100.0 | 1.8 | 7.54 | 0.15s |
| CP-SAT | **100.0** | **0.0** | **7.54** | 1.2s |

## H  MULTI-DEPENDENCY CRITICAL PATH EXTRACTION ALGORITHM

This section presents the Multi-Dependency Critical Path Extraction algorithm that extends traditional critical path methods to handle OR-AND prerequisite logic encoded in the Logic Adjacency Matrix (LAM). Traditional critical path algorithms assume singular path dependencies, but university course planning involves multiple, nested, and conditional prerequisite paths. Our algorithm correctly handles these complex dependency structures.

### H.1  ALGORITHM: MULTI-DEPENDENCY CRITICAL PATH DETECTION

**Input:**

- $G(V, E)$: Course dependency graph
- LAM: Logic Adjacency Matrix ($n \times n$)

- $D$: Dictionary mapping each course to its duration (e.g., number of weeks or terms)

**Output:**

- CriticalPaths: List of longest valid prerequisite paths (considering AND/OR logic)

**Procedure** MULTIDEPENDENCYCRITICALPATHDETECTION$(G, \text{LAM}, D)$:

**Step 1. Initialize:**

- inDegree$[v] \leftarrow 0$, EarliestStart$[v] \leftarrow 0$, Predecessors$[v] \leftarrow \emptyset$ for all $v \in V$
- CriticalPaths $\leftarrow \emptyset$

**Step 2. Parse LAM to build Multi-Dependency DAG:**

- For each column $j$ in LAM: Extract non-zero values cells$[j] = \{(i, \text{val}) \mid \text{LAM}[i][j] \neq 0\}$
- Group prerequisites by AND index: andGroups $=$ groupBy(val) from cells$[j]$
- For each group in andGroups, for each $(i, \text{val})$ in group:
  - Add edge $(i \rightarrow j)$ to $E$; inDegree$[j] \leftarrow$ inDegree$[j] + 1$
  - Record that $i$ is one valid prerequisite of $j$

**Step 3. Topological Sorting with OR-AND Validation:**

- Initialize Queue $\leftarrow \{v \in V \mid \text{inDegree}[v] = 0\}$
- While Queue $\neq \emptyset$: pop $u$ from Queue
- For each successor $v$ of $u$: if AND prerequisites of $v$ are satisfied, then:
  - EarliestStart$[v] \leftarrow \max(\text{EarliestStart}[v], \text{EarliestStart}[u] + D[u])$
  - inDegree$[v] \leftarrow$ inDegree$[v] - 1$
  - If inDegree$[v] = 0$, append $v$ to Queue

**Step 4. Backtrack to extract Critical Path(s):**

- maxLength $\leftarrow \max_v(\text{EarliestStart}[v] + D[v])$
- For each node $v$ such that EarliestStart$[v] + D[v] =$ maxLength:
  - Backtrack from $v$ to source(s) using Predecessors$[v]$; store path(s) in CriticalPaths

**Return** CriticalPaths

**Complexity analysis.** The algorithm runs in $O(|V| + |E|)$ time for graph traversal, with an additional $O(n^2)$ cost for parsing the LAM, where $n = |V|$. The LAM parsing is performed once offline per institution; the topological sort and backtracking are performed per student query. The algorithm correctly handles OR-AND logic by validating AND group satisfaction before updating earliest start times, ensuring that courses with multiple prerequisite options (OR logic) are scheduled only when at least one valid prerequisite path is satisfied, while courses with mandatory co-requisites (AND logic) wait for all prerequisites to complete.