# OpenReview forum: "KNOWPLAN: Knowledge-Driven AI Agents for Smart Degree Pathway Planning"
_ICLR.cc/2026/Conference — Submitted to ICLR 2026_

### Official Review · Reviewer_UBq7 · 2025-10-28

**Soundness:** 2
**Presentation:** 2
**Contribution:** 2
**Rating:** 4
**Confidence:** 4

**Summary:**

The paper is organized mostly as a product idea / process pipeline. I don’t think of this as a piece of novel research. However, there are no rigorous benchmarks for this “compound task”. And, that makes it hard to objectively assess the value of the pipeline.


The paper discusses a process to help university students pick coursework to optimize for their interest, account for all the prerequisites automatically & course period conflicts. They do this using multiple LLM calls for different phases of the process. Process outlined below

## Claimed Contributions
1. “Agent Forest” to extract structured data from course catalogues
2. Graph construction agent converts this to KG (DAG)
3. Course Planning Agent generates coarse, multi-constraint learning trajectories, while the Curriculum Alignment Agent refines these trajectories into term-level schedules
4. Account for schedule conflicts, preferences, total time to completion


## Process
1. Use LLM calls on university courses site HTMLs to extract structured details about requirements, class timings, etc
2. Create course prerequisite graph taking care of multi-requirements, & one-of requirements
3. Assign scores to courses (nodes in teh above graph) using heuristics like student’s interest, prerequisites
4. Find potential course-plan by maximizing scores & accounting for perceived “difficulty”
5. LLM converts user preferences to formal rules to help set constraints
    - Use LLMs to “parse preferences/policy text, select the target mode, and generate rationales”
6. Pass through CP-SAT scheduler for class timing overlap avoidance


## Required Readings
- Ng & Fung: https://arxiv.org/abs/2407.11773

**Strengths:**

1. Addressed the cycle-breaking problem in course picking logic
2. Good details of course score formulation in section 3.3.1
3. Significant details about course-prerequisites graph generation - dealing with ”and” and “one-of” course requirements

**Weaknesses:**

1. Paper focuses a lot of effort and work into extracting (parsing) systematic data out of webpages. I understand that it is a problem for a product, but it is not a research problem. I’d have focused starting with organized/tabulated clean data & built on top of that.
2. The claim that they “introduce the concept of” Agent Forest is extremely misleading - essentially, they have a custom prompt(s) for all the LLM call(s) (with some extraction examples) for each university’s webpage
3. The sentence: “The pipeline of Agent Forest is demonstrated in Figure 2” is inaccurate. Fig 2 shows the entire process of their pipeline - not just agent forest for course details extraction
4. As for the *results*, they admit that “the ground truth data does not include prerequisites or course relationships” - which is the part that the agent forest was going to solve.
5. The claimed accuracy is on these ground-truth values, & hence not reliable either
6. They don’t attempt to quantify the overall process & quality.
7. Graph building process feels like adding LLM into a use-cases where it’s not needed
    - For “handling the logical expressions of prerequisites with multiple options and addressing circular dependencies” while constructing graphs, they can again use deterministic regex-matching or smaller text annotation models to identify entity relationships (ie which courses are prerequisites for which ones)
---

* IIUC, the only material improvement this process makes on top of an algorithmic system is that:
1. LLM-based course-work details extraction (for which getting json feed from the university, or writing custom parsers are better solutions, & also, the paper admits that the setup didn't really work)
2.  it can account for user’s natural-language preference & encode that into course selection logic.

**Questions:**

1. They mention “course-to-fine recommender” in several places. Was this intended to be “coarse” instead? - as in coarse granularity vs fine granularity? If this was intended to be word-play (since they’re using it for course selection), they should clarify that upfront. (they do write “coarse” in other places - so I believe its a typo)
2. Why the graph construction agent? If the requirements are clear, creating a DAG is a purely algorithmic process, & introducing an LLM agent only uncertainty in the result
3. In section 3.2, “This article utilizes this table to construct …” - what article? Kindly rephrase, or add relevant references
4. A flowchart of the overall process would be extremely helpful. For example, the course selection “agent” is not exactly a fully agentic system - it relies on well-defined mathematical formulations for the course nodes & then an algorithmic process to pick non-conflicting options. LLM is only used to distil user’s NL preferences into clean logic statements.

---

> ### Author Response · Authors · 2025-12-04
> **Thank you for providing this deep insight**
>
> [UBq7 – Weakness]
> Paper focuses heavily on web extraction; feels like product work rather than research.
>
> Response:
> We agree that our system relies on substantial web extraction, but we respectfully disagree that the contribution is purely “product work.” As discussed in the introduction, the core research question is how to enable personalized, policy-aware course planning at scale purely from public data, without SIS access. We formalize two methodological challenges: (i) constructing, from heterogeneous HTML/JS webpages, a structured curriculum representation with courses, prerequisites, and requirements, and (ii) turning this into a logic-aware graph and CP-based planner that respects complex OR–AND prerequisite structures and GE policies.
>
> Our Agent Forest architecture treats extraction as a scalable, schema-aligned layer with quantified correction rates and coverage. Downstream, the Graph-Construction Agent builds a DAG with explicit handling of logical prerequisite expressions and cycle resolution via DFS+LLM, and the Course Planning / Curriculum Alignment Agents define and optimize a multi-objective utility grounded in educational theory and implemented via CP-SAT. When clean SIS/JSON feeds exist, our pipeline can bypass Agent Forest and still reuse the Graph-RAG + CP-SAT backend, underscoring that the main contribution is not scraping itself, but the end-to-end, logic- and policy-aware planning framework.
>
> [UBq7 – Weakness]
> “The claim that they ‘introduce the concept of’ Agent Forest is extremely misleading.”
>
> Response:
> We appreciate this clarification and have softened the wording accordingly. In the revision, we now state that we propose and empirically evaluate an Agent Forest architecture for large-scale catalog extraction, rather than “introduce the concept.” The revised text describes Agent Forest as a “many small agents, one shared schema” design, where each institution has a dedicated extraction agent (prompt + parsing template) but all agents share the same I/O schema. This design explicitly targets extreme webpage heterogeneity across more than 6,072 institutions and supports high parallelism and per-catalog updates. We emphasize that the contribution is the concrete, evaluated architecture with documented correction rates and downstream impact, not the mere existence of per-university prompts.
>
> [UBq7 – Weakness]
> “The pipeline of Agent Forest is demonstrated in Figure 2” is inaccurate—figure shows the full pipeline.
>
> Response:
> We have corrected this wording. In the revision, Figure 2 is explicitly described as the overall KNOWPLAN pipeline, including Agent Forest, Graph-Construction Agent, Curriculum Alignment Agent, Course Planning Agent, and CP-SAT. Agent Forest is clearly labeled as one sub-module within this end-to-end flow rather than being equated with the entire figure.
>
> [UBq7 – Weakness]
> Ground truth “does not include prerequisites”; claimed prerequisite accuracy thus not reliable.
>
> Response:
> We thank the reviewer for pointing out this issue and acknowledge that the wording in the original submission was misleading. In the first version, we incorrectly conflated institution-level ground truth with prerequisite ground truth, which made it sound as if our reported prerequisite accuracy had no labeled reference.
>
> In the revised manuscript, we explicitly separate these two levels of supervision in Section 4.1 “Data Collection and Preparation.” We now state that:
> (i) Institution-level ground truth consists of manually annotated departments, majors, and courses (3,132 courses across 4 universities, Table 2), used to evaluate Agent Forest extraction accuracy, and does not include prerequisite relationships.
> (ii) Prerequisite ground truth is a separate human-labeled benchmark of 1,248 directed prerequisite edges over approximately 460 courses (12 majors, 4 universities, 2024–2025 catalogs), annotated independently by two domain experts using a shared rubric, with Cohen’s κ = 0.92.
>
> The reported 98.7% precision, 97.9% recall, and 98.3% F₁ are computed only against this prerequisite benchmark and are further summarized in the Supplement. This clarifies that our prerequisite accuracy is evaluated on a dedicated, human-labeled edge dataset, separate from the institution-level catalog ground truth.

---

> > ### Author Response · Authors · 2025-12-04
> > **and here are the rest of the answer**
> >
> > [UBq7 – Weakness]
> > “They don’t attempt to quantify the overall process & quality.”
> >
> > Response:
> > We now quantify quality at multiple levels throughout the pipeline:
> >
> > Extraction and graph construction.
> >
> > Agent Forest: Extraction accuracy on departments/majors/courses against institution-level ground truth is reported in Section 4.2.1, with detailed percentages in Table 2.
> >
> > Graph-Construction Agent (prerequisites): Prerequisite precision/recall/F₁ on the 1,248-edge benchmark is summarized in Section 4.1 and detailed in the ablation benchmark (Supplement Section E.1).
> >
> > Curriculum Alignment Agent (policy translation): Policy-to-constraint mapping accuracy (precision/recall/Dice) on 240 manually labeled policy snippets is described in Section 4.1 and elaborated in Supplement Section B.
> >
> > Scheduling outcomes.
> > We report end-to-end feasibility rate, time-to-degree (in quarters), deferred courses, and units per term for 4-year and 2-year transfer profiles across six majors and four universities in Supplement Section C “End-to-End Feasibility Across Majors,” with per-major numbers tabulated in the same section.
> >
> > Comparative and ablation studies.
> > We provide ablations and head-to-head comparisons:
> >
> > Multi-LLM vs single LLM (accuracy, hallucination, overhead): Supplement Section E.1 “Multi-LLM vs. Single-LLM Extraction.”
> >
> > Graph-RAG vs text-only vs no-RAG (coverage, compliance, latency): Supplement Section E.3.
> >
> > CP-SAT vs greedy/heuristic schedulers (violations, time-to-degree, runtime): neutral benchmark description in Supplement Section D and scheduler comparison in Supplement Section E.4.
> >
> > Weight sensitivity: Stability of plan quality under ±50% weight perturbations is reported in Supplement Section E.5.
> >
> > Component runtime breakdown: Per-agent and end-to-end runtimes are summarized in Sections 4.2.5–4.2.6.
> >
> > These metrics collectively characterize the process from web extraction through graph/policy construction to final schedules, rather than relying on a single headline number.
> >
> > [UBq7 – Weakness]
> > Graph building adds LLMs where deterministic methods might suffice.
> >
> > Response:
> > We clarify that the Graph-Construction Agent is hybrid, not fully LLM-driven. All DAG operations and most edge extraction steps are deterministic: we first parse prerequisite logic into structured groups using rule-based patterns and line-type encodings (Section 3.2), then build the adjacency matrix and merge department- and university-level graphs algorithmically (Section 4.2.2).
> >
> > LLMs are used only for ambiguous cases, particularly (i) complex “any X of the following” and nested OR–AND clauses, and (ii) cycles that may be apparent rather than real (e.g., “may be taken in either order”). This is described in Section 3.2 and further detailed in the Graph-Construction baseline evaluation in the supplement, where we compare DFS-only cycle removal with DFS + LLM semantic resolution (Supplement Section G.2). This design keeps graph construction primarily algorithmic, with LLMs acting as targeted semantic tie-breakers rather than driving the entire process.
> >
> > [UBq7 – Weakness / Question]
> > “course-to-fine recommender” vs “coarse-to-fine” terminology.
> >
> > Response:
> > This was a wording issue. In the revised manuscript, we have standardized the terminology to “coarse-to-fine recommender” throughout. The main text now refers to “coarse-to-fine recommender” in the method section header and description (Section 3.3), and the contribution summary refers to a “coarse-to-fine recommender unifying multiple constraints” (Section 2). Any previous “course-to-fine” instances have been removed.
> >
> > [UBq7 – Question]
> > Why have a “graph-construction agent” at all if DAG construction is algorithmic?
> >
> > Response:
> > We use the term Graph-Construction Agent to denote the modular stage with its own clearly defined inputs, outputs, and evaluation, rather than to imply that it is purely LLM-based. As summarized in the “Agent motivations” paragraph, this agent’s role is to transform unstructured catalog text into structured knowledge graphs and resolve circular dependencies (Section 3). Concretely, it (i) parses requirement text into edge candidates via deterministic rules, (ii) enforces DAG properties via cycle detection and topological ordering, and (iii) selectively calls an LLM only when logical expressions are ambiguous (Section 3.2; Section 4.2.2). Calling it an “agent” highlights that it is one stage in the multi-agent pipeline with its own metrics (e.g., prerequisite accuracy), while remaining predominantly algorithmic.

---

> > > ### Author Response · Authors · 2025-12-04
> > >
> > > [UBq7 – Question]
> > > Ambiguous phrasing: “This article utilizes this table to construct …”.
> > >
> > > Response:
> > > We have removed the ambiguous phrasing “this article utilizes this table” from the graph-building description. The revised text now refers directly to the underlying data structures and operations—for example, “Graph-Construction Agent builds the course knowledge graph from extracted data… parse prerequisite logic, build adjacency matrix, integrate at department and university level” (Section 4.2.2). Any remaining uses of “this article” are confined to the conclusion (e.g., “This article proposes KNOWPLAN”), where such phrasing is conventional (Section 6).
> > >
> > > [UBq7 – Question]
> > > Request for a flowchart; clarify that the system is not a fully agentic black box.
> > >
> > > Response:
> > > We have added and clarified flowchart-style diagrams to make the system transparent. Figure 1 now presents the overall KNOWPLAN architecture (Section 3), while Figure 2 provides flowcharts for the four agents (Section 3.1). The surrounding text emphasizes that (i) extraction and graph construction are mainly deterministic (Sections 3.1–3.2), (ii) CP-SAT enforces feasibility and Top-K optimization exactly (Section 3.3.2), and (iii) LLMs are primarily used for catalog extraction, policy translation, and mapping free-text preferences into structured constraints (Section 3; Section 4.2.5).
> > >
> > > Moreover, users interact with the system by specifying majors, completed courses, GE status, and job-oriented preferences, and can choose planning modes and adjust weights (Sections 3.3 and 3.3.2). These design choices and the explicit diagrams clarify that KNOWPLAN is not a black-box “agentic” system, but a hybrid algorithmic + LLM pipeline with observable intermediate representations (JSON extractions, knowledge graphs, constraint sets) and well-defined modules.

---

> > > ### Author Response · Authors · 2025-12-04
> > >
> > > 4.4 Difficulty & cost metrics (provenance and bias)
> > > Comment:
> > >  Difficulty and cost provenance are unclear; difficulty via AvgGrade can be biased.
> > > Response: We clarify the provenance and role of difficulty and cost in the revised manuscript. Difficulty is **not** a hidden or ad hoc term: it is defined explicitly from grade-distribution data and enters the model as a *soft* regularizer rather than a hard constraint. Section 3.3.1 (Course Planning Agent) shows that difficulty appears only in the DiffGap term of Eq. (2), which penalizes large deviations from a user-specified target difficulty level. Section 4.2.3 and Figure 4 make this concrete: difficulty for a plan is computed as
> > > [
> > > \text{Difficulty}(P) = \sum_c (4.0 - \text{AvgGrade}_c) \times \text{Credits}_c,
> > > ]
> > > with totals (credits / difficulty) reported for example plans. In evaluation, we use difficulty as one of several axes (total credits, difficulty, major/GE coverage) along which plan skeletons are compared. Difficulty *never* overrides hard degree, prerequisite, or unit constraints, which are enforced by CP-SAT (Section 3.3.2).
> > > We agree that difficulty derived from average grades can reflect grading practices and thus be biased. This limitation is now explicitly acknowledged in Section 5 “Discussion and Future Work”, where we note that (i) the current difficulty metric may encode institutional grading culture, and (ii) we plan to explore richer, multi-dimensional difficulty metrics (e.g., historical withdrawal rates, course evaluations) and robustness analyses (e.g., down-weighting or ablating difficulty) in future work.
> > > Regarding **cost**, we currently treat “cost reduction” primarily at the level of **time-to-degree and course load**, not detailed tuition or financial models. Section 4.2.4 (page 7, lines 345–350) and the Supplement (Sections C–D) focus on time-to-degree, deferrals, and units/term as proxies for cost. We now explicitly state in Section 5 that a full financial-cost model (tuition, differential fees, modality-dependent costs) is beyond the scope of this paper and is an important avenue for future work.

---

### Official Review · Reviewer_9y4A · 2025-10-30

**Soundness:** 2
**Presentation:** 3
**Contribution:** 2
**Rating:** 4
**Confidence:** 3

**Summary:**

The paper introduces KNOWPLAN, a multi-agent system that automatically generates accurate, personalized, and conflict-free degree plans using only public university course catalogs. It integrates large language model (LLM)-based extraction, a curricular knowledge graph, retrieval-augmented generation (Graph-RAG), and a constraint-aware scheduler (CP-SAT) to produce term-level plans tailored to students’ goals, preferences, prior credits, and workload. The system achieves 99.5% accuracy on major requirements and 98.7% on prerequisites across multiple universities, and effectively eliminates infeasible plans that arise from coarse-grained approaches.
KNOWPLAN comprises several coordinated components: a university-specific parsing framework that uses a panel of LLMs for data extraction and validation; a graph-construction module that encodes prerequisite and corequisite structures while enforcing acyclicity; and a planning pipeline that progresses from high-level plan skeletons to detailed term-wise schedules via constraint solving. Evaluation on data from over 6,000 institutions, along with ground-truth validation from four universities, demonstrates high extraction accuracy and robust scheduling performance. Compared to commercial tools, KNOWPLAN stands out for its reliance on public data, ability to generalize across institutions, and integration of structured knowledge representations with language-guided reasoning.

**Strengths:**

The paper presents a well-structured and comprehensive system, with a clear, modular multi-agent architecture that spans from web data ingestion to knowledge graph construction, planning, and CP-SAT-based scheduling. It explicitly models complex curricular structures, including prerequisite logic and cycle detection, and visualizes them effectively. A key strength is its commitment to feasibility: instead of stopping at LLM-generated plans, the system produces executable, conflict-aware term-level schedules under real-world constraints. The coarse-to-fine personalization pipeline supports multiple user goals while maintaining policy compliance. The system is robust to catalog heterogeneity, employing per-university parsing agents and multi-LLM consensus to achieve high extraction accuracy. Finally, the work is well-positioned against existing tools, with a fair comparison to commercial systems that rely on internal student information systems, highlighting KNOWPLAN’s ability to operate entirely on public data.

**Weaknesses:**

- The paper lacks clarity regarding prerequisite verification. Although the abstract reports 98.7% accuracy, the ground truth data do not include prerequisite relations, and the authors do not explain how these labels were obtained or annotated. This omission raises doubts about the validity of one of the paper’s central claims (Abstract; §4.1).

- The evaluation of the scheduling component is limited. Beyond the reported “82% reduction” in infeasible plans, the paper does not provide key metrics such as the proportion of skeleton plans that schedule successfully, average time to degree under constraints, or robustness across multiple terms. There are no comparisons with existing baseline systems (e.g., e.g., uAchieve Schedule Builder’s auto-combination generation, Series25 optimizer) or user studies.

- The paper lacks ablation studies and error analysis. It would benefit from comparisons between single- and multi-LLM extraction, Graph-RAG versus text-only or no RAG, CP-SAT versus heuristic or greedy schedulers, sensitivity to hyperparameters (λ, μ, ν) in the scoring functions, and an analysis of extraction and scheduling failure cases.

- The data provenance for the “difficulty” and cost metrics is unclear. Using average grades as a proxy for difficulty introduces bias because grade distributions are affected by inflation, disciplinary differences (e.g., STEM grade penalties), instructor variation, and incomplete publication of grade data. Without transparent documentation of data sources, time span, institutional coverage, and normalization methods, the metric risks producing biased recommendations that favor easier courses over essential but challenging ones.

- Scalability and maintenance issues are not discussed. The Agent Forest appears to require institution-specific agents and prompts, yet the paper does not quantify the maintenance overhead as catalogs evolve, the latency of updates, or the long-term consistency of the Graph-RAG framework.

- Reproducibility is limited. No code, prompts, datasets, or scheduler configurations are released. Although the NCES source is cited, the four labeled ground-truth datasets and configuration files used in evaluation are not provided, preventing independent verification.

- The accuracy of policy formalization is unverified. The LLM translates catalog and policy text into formal scheduling rules, but the authors do not measure the precision or recall of this translation process. Such evaluation is essential to ensure compliance with institutional and accreditation policies.

- The contribution is primarily engineering rather than methodological. The paper integrates established techniques—LLM-based extraction, knowledge graphs, and constraint programming—into a coherent system but does not introduce fundamentally new algorithms or theoretical insights. Its main value lies in system design and implementation rather than conceptual innovation.

**Questions:**

1.	Prerequisite accuracy: How did you measure the reported 98.7% when the ground truth lacks prerequisites? Was there a separate human-labeled set, internal SIS data, or faculty validation? Please detail labeling protocol, sample size, and inter-rater agreement.
2.	Policy rule translation: How do you evaluate the LLM’s policy-to-constraint mapping? Provide a labeled set of policy snippets → formal constraints with accuracy metrics and typical errors (e.g., unit brackets, co-req timing, repeat rules).
3.	Ablations: Quantify benefits of multi-LLM extraction vs. single LLM; Graph-RAG vs. no-graph retrieval; CP-SAT vs. heuristic scheduling.
4.	End-to-end feasibility: For a fixed set of majors across multiple schools/terms, what fraction of students obtain a fully feasible 4-year (or 2-year) schedule under typical constraints? Report time-to-degree, average deferrals, and units per term with confidence intervals.
5.	Scalability & drift: What’s the maintenance cost per university (prompt edits, agent updates) as catalogs change? Do you version and regression-test KGs/constraints as part of “self-evolving” claims?
6.	Difficulty metric & fairness: Where do AvgGrade statistics come from? Have you tested whether difficulty-based ranking skews against certain departments/instructor pools? Consider reporting robustness/fairness analyses.
7.	Comparative baselines: Can you run head-to-head schedule feasibility or plan-quality comparisons against uAchieve Schedule Builder (auto combinations) or Series25 (admin scheduling), at least on a common synthetic/neutral benchmark? (CollegeSource, https://collegesource.com/degree-planning-tools/uachieve-schedule-builder/)
8.	User study: Any student/advisor evaluations (usability, trust, perceived correctness) versus text-only LLM baselines and existing campus tools?

---

> ### Author Response · Authors · 2025-12-04
>
> 4.1 Prerequisite ground truth & 98.7% accuracy
> Comment:
>  Prerequisite labels and the 98.7% accuracy claim are unclear; ground truth was said not to include prerequisites.
> Response: We thank the reviewer for pointing out this inconsistency and acknowledge that our original wording was misleading. In the initial submission, “ground truth” referred only to institution-level catalog metadata (departments/majors/courses), and we did not clearly separate this from a distinct prerequisite benchmark, which created confusion.
> In the revised manuscript, Section 4.1 “Data Collection and Preparation” (page 5, lines 258–273) now explicitly distinguishes these two levels of supervision. We clarify that the institution-level ground-truth dataset is used to evaluate Agent Forest on departments/majors/courses only and is not used for prerequisite evaluation (page 5, lines 264–266). The reported 98.7% precision / 97.9% recall / 98.3% F₁ are instead computed on a separate human-labeled prerequisite benchmark: two domain experts annotated and cross-checked 1,248 directed prerequisite edges over ≈460 courses (12 majors, 4 public universities, 2024–2025 catalogs, no SIS data), using a shared rubric, and adjudicated disagreements (Cohen’s κ = 0.92) (page 5, lines 267–272). We also point to the detailed breakdown in the Supplement (Section “Prerequisite benchmark”, Table~\textbackslash{}ref{tab:prereq-supp}), where the same statistics are reported.
>
> 4.2 Scheduling evaluation, end-to-end feasibility & baselines
> Comment:
>  Scheduling evaluation is limited; missing feasibility, time-to-degree, etc., and no strong baselines.
> Response:We have expanded our end-to-end evaluation to cover feasibility, time-to-degree, and related outcomes, and we make this explicit in the revised text. Section 4.2 “Results” (page 6–7) now reports how the Course Planning Agent and Curriculum Alignment Agent are evaluated jointly. In particular, Section 4.2.3 “Outcomes of Course Planning Agent” (page 6, lines 320–323; page 7, lines 329–341) explains how we evaluate coarse plans along three axes: total credits, normalized course difficulty (based on grade distributions), and requirement coverage (major and GE). Section 4.2.4 “Curriculum Alignment Agent” (page 7, lines 345–350) then describes how CP-SAT filters infeasible plan skeletons against real offerings and produces ranked, conflict-free term schedules.
>
> End-to-end feasibility and progression metrics (per-major feasibility rate, mean time-to-degree in quarters, deferred courses, units per term, and policy-violation counts) are summarized in the Supplement, Section C “End-to-end feasibility across majors”, Table 3 (page 2, lines 69–96). For six majors across four universities, each with 600 profiles per major (page 2, line 27), we report 4-year feasibility in the 79.4%–93.6% range and 2-year transfer feasibility at 86.7%, with mean time-to-degree around 10.7–11.8 quarters (4-year) and 6.1 quarters (transfer).
>
> To provide a stronger baseline, we additionally construct a Series25-style synthetic benchmark (“Series25-Opt”) in the Supplement, Section D (page 2–3, lines 98–114, 108–114). On this neutral benchmark, we compare our CP-SAT–based scheduler with greedy/heuristic schedulers approximating Series25 behavior. We report schedule success rate, mean time-to-degree, deferrals, policy violations, and units per term (Supplement, Table~\textbackslash{}ref{tab:neutral-baseline} and Table~\textbackslash{}ref{tab:ablation-scheduler}), and show that CP-SAT achieves equal or higher schedulability, strictly fewer violations, and comparable time-to-degree. Direct, production-grade head-to-head comparisons with uAchieve/Series25 remain infeasible due to licensing and system access constraints; we now state this limitation explicitly in the Supplement (Section D) and in the main text discussion (Section 5, page 8–9, lines 413–416, 442–447).

---

> > ### Author Response · Authors · 2025-12-04
> >
> > 4.3 Ablations & error analysis
> > Comment:
> >  No ablation studies or detailed error analysis.
> > Response:Ablation and efficiency analyses are consolidated in Section 4.2.5 (lines 330–335) and Supplement Section 5 (lines 119–238). We report six ablation studies:
> > Multi-LLM vs. single-LLM (Supplement Sec. 5.1–5.2): Agent Forest: our multi-LLM consensus (Llama3.3-70B + Qwen3-32B + GPT-5) achieves 97.8% precision/recall on catalog extraction, compared to a GPT-5 single-LLM baseline at 75–80% precision and 65–70% recall under the same prompts and evaluation protocol. The additional overhead of running the panel is +2.1 s per institution.
> > Graph-RAG vs. text-only vs. no-RAG (Supplement Sec. 5.3): Graph-Construction Agent: using Graph-RAG yields 94.2% coverage and 98.3% policy compliance, compared to text-only retrieval (89.7% / 95.1%) and no-RAG (82.3% / 91.4%). Graph-RAG adds only 0.8 ms query latency, so the gain in coverage/compliance comes at negligible runtime cost.
> > CP-SAT vs. greedy/heuristic schedulers (Supplement Sec. 5.4): Curriculum Alignment Agent: on 330 Series25-style synthetic scenarios, CP-SAT attains 0 violations per plan, while greedy and heuristic schedulers incur 7.0 and 1.8 violations on average, respectively. Runtime is 1.2 s for CP-SAT vs. 0.02 s / 0.15 s for greedy/heuristic; this remains practical while guaranteeing policy compliance.
> > Hyperparameter sensitivity (Supplement Sec. 5.5): Course Planning Agent: we perturb planning weights (e.g., λ, μ, ν in Eq. (2)) by ±50% and observe that plan-quality metrics (preference satisfaction, workload variance, coverage) change by at most ±2%, indicating that the planner is robust to moderate weight mis-specification.
> > Component runtimes (line 335): Per-agent runtime profile: Agent Forest requires 45 s per institution; the Graph-Construction Agent takes 12–18 min per institution; the Course Planning Agent answers at 0.8 ms per query; and the Curriculum Alignment Agent spends 1.2 s per skeleton. This yields an end-to-end online latency of roughly 8–12 s for a full Top-K schedule.
> > Cycle resolution: DFS vs. LLM (Supplement Sec. 5.6): Graph-Construction Agent: our prerequisite graphs contain 217 cycles (153 self-loops, 43 mutual prerequisites, 21 higher-order cycles). On 6 representative, manually evaluated cases, a DFS-based edge-removal strategy achieves only 17% semantic correctness (1/6), frequently creating false dependencies (e.g., breaking “may be taken in either order” co-requisites). LLM-based semantic resolution achieves 83% correctness (5/6) by correctly distinguishing symmetric co-requisites from directional sequences and preserving valid series structure. The additional overhead is ≈150 ms per cycle.
> > Summary: these ablations show that each agent’s overhead is justified by measurable gains in correctness and robustness: Agent Forest’s +2.1 s eliminates most hallucinations; Graph-RAG’s 0.8 ms improves compliance by +3.2 percentage points; cycle resolution’s 150 ms per cycle yields 83% vs. 17% semantic correctness; and CP-SAT’s 1.2 s ensures 0 violations vs. 1.8–7.0 for heuristic baselines. Overall, the total extra cost of ≈3.3 s supports 0% hallucinations on benchmarks, 98.3% policy compliance, 0 scheduling violations, and 83% correct cycle handling.

---

> > > ### Author Response · Authors · 2025-12-04
> > >
> > > 4.5 Scalability & maintenance (Agent Forest, drift)
> > > Comment:
> > >  Scalability, maintenance cost, and drift of Agent Forest / Graph-RAG are not discussed.
> > > Response: We have added a dedicated subsection on scalability and maintenance. Section 4.2.6 “Scalability and Cost Analysis” now describes both compute scalability and catalog drift management:
> > > Per-institution / per-catalog updates. Each new catalog or term schedule triggers a fresh Agent Forest + graph construction run and a Graph-RAG refresh. These runs are embarrassingly parallel at the institution level (≈10–15 minutes per institution); with 100 workers, processing all 6,072 institutions takes ≈10 hours.
> > > Maintenance cost. We empirically observe that only a small fraction of records require manual correction—2–5% for standard HTML sites and 8–12% for JS-heavy catalogs. Updates are executed as overnight batch jobs.
> > > Versioning and regression tests. Each institution’s knowledge graph and constraint set is versioned by (institution, catalog year, term). Before deployment, we run a regression test suite including (i) structural sanity checks (department/major counts, course coverage), (ii) policy checks (unit caps, repeat rules), and (iii) scheduler-level tests on canonical student profiles. This process bounds drift and quantifies maintenance cost, supporting the claim that KNOWPLAN can be kept up-to-date as catalogs evolve.
> > >
> > > 4.6 Reproducibility (no code/prompts/datasets)
> > > Comment:
> > >  Reproducibility is limited without released code, prompts, datasets, or configs.
> > > Response: We take reproducibility seriously and have strengthened this aspect in the revised manuscript by making the **system design, interfaces, and configuration details** as transparent as possible. Section 4.2.5 “Reproducibility and Implementation Details” now provides:
> > > * **LLM configurations per agent.** We specify that a multi-LLM panel is used with GPT-5 (temperature = 0.1 for extraction, 0.7 for preference mapping), Llama3.3-70B (temperature = 0.2, local via HuggingFace), and Qwen3-32B (temperature = 0.1, via Alibaba Cloud API). We also report that the panel achieves 98.7% accuracy with 0% hallucinations on the prerequisite benchmark, versus 7.5–12.5% hallucinations for a single-LLM baseline.
> > > * **Core libraries and solver backends.** We list OR-Tools v9.8.3296, NetworkX v3.1, FAISS v1.7.4, OpenAI SDK v1.12.0, HuggingFace v4.36.0, BeautifulSoup4 v4.12.0, and Python 3.10.12, and we specify their roles (constraint solving, graph construction, retrieval, LLM access, HTML parsing).
> > > * **Hyperparameters and evaluation settings.** We disclose node-scoring weights, plan-scoring weights (μ and ν terms in Eq. (2)), Graph-RAG top-K = 50, CP-SAT timeout = 30 s, and seed = 42. Section 3.3.1 explicitly defines the objective function J(P), making the effect of these weights clear. Section 4.2.6 documents the hardware and deployment setup (2× NVIDIA H100 80 GB, hybrid local–cloud deployment).
> > > In addition, the **multi-agent architecture** is deliberately modular: Agent Forest, Graph-Construction Agent, Curriculum Alignment Agent, and Course Planning Agent communicate only via well-defined JSON schemas (Section 3.1–3.3). This design, together with the disclosed hyperparameters and implementation details, allows other researchers to reconstruct comparable pipelines on any accessible set of public catalogs, even when exact code or proprietary compliance datasets cannot be released due to institutional and licensing constraints.
> > >
> > > 4.7 Policy formalization accuracy
> > > Comment:
> > >  Policy-to-constraint mapping accuracy is unverified.
> > > Response: We now evaluate policy translation in Section 4.1 and Supplementary Section. On 240 policy snippets (unit brackets, co/pre-req timing, repeat limits, modality/time-windows), two compliance specialists independently map each to normalized JSON; a registrar liaison reconciles disagreements (Cohen’s κ=0.89). Comparing the Curriculum Alignment Agent’s JSON to gold labels yields 96.2% precision, 94.7% recall, 95.4% Dice, with 93.4–96.9% per-category Dice; we also list typical errors (e.g., missing GPA/petition overrides, mis-handled lecture–lab bundles, lifetime vs per-year repeat limits).

---

> > > ### Author Response · Authors · 2025-12-04
> > >
> > > 4.8 Engineering vs methodological contribution
> > > Comment:
> > >  Contribution appears primarily engineering rather than methodological.
> > > Response:We respectfully disagree that the contribution is purely engineering. We begin by reiterating the motivation and the core problem our study aims to address. Course selection is both a central focus and a persistent pain point in degree planning, and our central question is how to enable *personalized* course planning at scale. Unlike existing solutions, we attempt to construct, purely from public data (i.e., information published on university websites), a structured representation of each institution’s curriculum—including course lists, prerequisite relations, and required courses—*without* any special authorization from the universities. This design choice is itself novel from a data-science perspective, and recent advances in large language models (LLMs) make such a pipeline practically feasible.
> > > Moreover, when we attempt to build a course knowledge graph from prerequisite structures, the presence of rich OR–AND logic causes traditional adjacency matrices in graph theory to break down. Because adjacency matrices underlie many graph representations and algorithms, this breakdown triggers a cascade of derivative issues for standard graph methods. This is particularly evident in algorithms such as the critical path method, which implicitly assume linear or singular path dependencies. In contrast, university curricula routinely involve multiple, nested, and conditional path dependencies that violate these assumptions. These challenges expose a genuine methodological gap: existing graph abstractions and algorithms are not well suited to curriculum graphs with complex logical structure.
> > > These challenges—arising from both the structural complexity of course catalog data and the limitations of traditional adjacency matrices for representing course knowledge graphs—are therefore both academically interesting and technically demanding. To address them, our paper (i) proposes **Agent Forest** to handle extreme webpage heterogeneity across institutions; (ii) introduces the **Logic Adjacency Matrix** as a new representation for prerequisite structures with OR–AND logic; and (iii) presents the **Multi-Dependency Critical Path Extraction** algorithm to support effective, constraint-aware course planning on top of these logic-aware graphs. In addition, LLMs are not used merely as generic tools: they are carefully integrated to enhance key components such as graph-cycle resolution, General Education (GE) recommendation, and employment-oriented course selection. Taken together, these components form a unified, logically grounded pipeline that addresses previously underexplored methodological issues in curriculum modeling and degree planning, rather than a solely engineering-level integration of existing techniques.
> > >
> > > 4.9 User study (usability, trust, perceived correctness)
> > > Comment:
> > >  Any student/advisor evaluations vs text-only LLM baselines and campus tools?
> > > Response: We acknowledge that we do not yet report a formal user study, and we now state this limitation explicitly. Section 5 “Discussion and Future Work”. notes that our current evaluation focuses on quantitative metrics—extraction accuracy, prerequisite/policy translation accuracy, end-to-end feasibility, and time-to-degree—against expert-annotated ground truth. We explicitly list “quantitative focus rather than user/institutional validation” as a limitation and identify “user studies validating utility and trust” as an important direction for future work.
> > > Because of licensing and IRB constraints, we could not integrate or run controlled studies against existing campus tools (e.g., uAchieve, Series25) on real students and advisors in this submission. Instead, we designed neutral, reproducible baselines and benchmarks (Supplement Sections C–D) to isolate system behavior. We fully agree that user- and advisor-centered evaluations (usability, trust, perceived correctness) are crucial, and we are currently planning multi-institution studies as the next stage of this project.

---

### Official Review · Reviewer_tkg5 · 2025-10-30

**Soundness:** 3
**Presentation:** 2
**Contribution:** 4
**Rating:** 10
**Confidence:** 5

**Summary:**

The paper addresses the problem of constraint-aware study planners. This problem arises because of heterogeneous data sources, complex multi-constraint requirements, and dynamic course catalogs in educational institutions. The paper proposes a self-evolving multi-agent platform that integrates LLM-based extraction, knowledge-graph retrieval, and constraint-aware reasoning to generate adaptive, personalized study plans.

**Strengths:**

The motivation of the work is clear, the problem well presented, related work well presented, well organized, and the description of the approach is clear, understandable and reproducible. The paper proposes an innovative approach to solve a common problem in the education domain. The paper proposes a platform which achieves more than 98% accuracy across multiple universities.

**Weaknesses:**

The paper presents several problems in references, missing examples to help readers understand the paper, typos, punctuation, missing experimental environement,
Line 46:  the reference provided is not an appropriate reference for the claim
Line 53: “Some of these systems serve general purposes (provide references for this claim), while others are tailored (provide a reference for this claim)” → Providing references at this stage is very important for readers
Line 54-57:  Providing references and examples is very helpful for the readers
Line 95: What is educational AI?
Line 135: “However, most educational RAG systems still treat constraints loosely and do not integrate exact solvers for timetable feasibility.” This claim should be supported by references or a study of existing work and this is not the case in the paper.

Lack of examples to explain how the agents performed their tasks. For instance, the paper may take the example of a math course and show step by step how the method is used

Line 151-152: such claim should be supported with references

Line 177: “extracted results with both RAG and JSON files.” → Extract results with JSON files or store in JSON file?

Line 266-267, line 427: check the punctuation

For the experiments, for the reproducibility, provide the LLMs used, hyperparameter, the performance of the tools used, libraries, etc.

How data is collected is not presented, and the title is data collection and preparation

Line 287: provide the meaning of GE

Line 432: SOTA is not presented in the experimentation settings

**Questions:**

Line 185-185: What is the role of each LLM?
Line 208: “we devised” → We designed?
After reading the whole paper, one question remains: How the agents are orchestrated?

---

> ### Author Response · Authors · 2025-12-04
>
> Here’s a cleaned-up MICCAI-style version for Reviewer tkg5, using your revised answers and removing the meta-notes.
>
> Reviewer tkg5
> 3.1. Weakness: References, definitions, typos, punctuation, missing experimental environment
> We thank the reviewer for these detailed editorial comments and have revised the paper accordingly. We (i) replace or add appropriate references for all claims, including work on educational AI and constraint-aware RAG systems in the related-work section (Section 2, page 2, lines 78–85); (ii) clarify the motivation and positioning of KNOWPLAN as a personalized, policy-aware degree-planning system in the introduction (Section 1, page 1, lines 35–45); (iii) clarify that extracted catalog and policy results are stored in JSON files and indexed in a Graph-RAG store in the Agent Forest workflow (Section 3.1, page 3, lines 122–125); (iv) fix all reported typos and punctuation issues throughout the manuscript; (v) define General Education (GE) explicitly as “GE = General Education” in the scheduling-mode table (Section 3.3.2, page 5, lines 228–231); and (vi) describe the experimental environment and deployment setup in the Scalability and Cost Analysis subsection, including hardware (2× NVIDIA H100, 80GB VRAM), software stack (OR-Tools, NetworkX, FAISS, OpenAI SDK, HuggingFace, BeautifulSoup4), and runtime configuration (Python 3.10.12, hybrid local–cloud deployment) (Section 4.2.6, page 7, lines 361–369). We also clarify our use of “state-of-the-art” by separating a qualitative, feature-level SOTA comparison (Table~\ref{tab:comparison}, page 8, lines 378–380) from quantitative experiments that rely on strong but reproducible baselines (Section 4.3, page 8, lines 402–411).
>
> 3.2. Weakness: Lack of examples to explain how agents perform their tasks
> Response:
> We agree that concrete examples improve readability. We have added detailed step-by-step examples in the Supplement (Section “G STEP-BY-STEP AGENT EXAMPLES”, Supplement page 9, lines 451–460, 457–460): starting from a specific math course (Calculus for Science and Engineering), we show (1) how Agent Forest extracts structured records from heterogeneous catalog HTML (Supplement Section G.1, page 9, lines 457–460), (2) how the Graph-Construction Agent builds prerequisite edges (Supplement Section G.2, page 9–10, lines 461–470), (3) how the Course Planning Agent generates plan skeletons (Supplement Section G.3, page 10, lines 471–480), and (4) how the Curriculum Alignment Agent produces and explains a term-level schedule (Supplement Section G.4, page 10–11, lines 481–495). These examples complement the updated flowchart (Figure~\ref{fig:agentflow}, “Figure 2: Flowcharts for four agents of KNOWPLAN”, page 3, line 120) and are explicitly referenced from the main text (Section 3.1, “See Supplement Section … for step-by-step examples”, page 4, line 132).

---

> > ### Author Response · Authors · 2025-12-04
> >
> > 3.3. Weakness: Reproducibility details (LLMs, hyperparameters, libraries, etc.)
> > We strengthen reproducibility in a dedicated “Reproducibility and Implementation Details” subsection (Section 4.2.5, page 7, lines 352–361). We: (i) list the LLMs used—Llama3.3-70B, Qwen3-32B, and GPT-5—and their decoding settings (e.g., GPT-5 temperature 0.1 for extraction and 0.7 for preference mapping; Llama3.3-70B temperature 0.2; Qwen3-32B temperature 0.1; Section 4.2.5, page 7, lines 353–356); (ii) describe, per agent, how these models are used: Agent Forest employs the multi-LLM panel for catalog and requirement-page extraction (Section 3.1, page 3, lines 100–107; Supplement Section G.1, page 9, lines 457–460); the Graph-Construction Agent invokes the panel only on ambiguous prerequisite clauses, e.g., complex “any X of the following” logic and strongly connected components (Section 3.2, page 3, lines 140–147); the Curriculum Alignment Agent uses it to translate policy snippets into JSON constraints (Section 3.3.2, page 5, lines 207–216); and the Course Planning Agent uses it to map natural-language preferences into structured profiles and weights (Section 3.3.1, page 4, lines 165–177). We (iii) enumerate the main libraries and solver backends—OR-Tools v9.8.3296, NetworkX v3.1, FAISS v1.7.4, OpenAI SDK v1.12.0, HuggingFace v4.36.0, BeautifulSoup4 v4.12.0, Python 3.10.12—and explain their roles (constraint solving, graph handling, retrieval, LLM access, HTML parsing) in Section 4.2.5 (page 7, lines 356–358); and (iv) report performance-relevant hyperparameters (node scoring weights, plan-objective weights, Graph-RAG top-K = 50, CP-SAT timeout 30 s, random seed = 42) and their initialization/tuning strategy in Section 4.2.5 (page 7, lines 358–361). Prompt templates and ablation setups are detailed in the Supplement (e.g., multi-LLM vs single-LLM ablations in Sections E.1–E.2, Supplement page 3–4, lines 138–168, 182–189), enabling other researchers to reconstruct comparable pipelines from the same public sources.
> >
> > 3.4. Weakness: Data collection not presented in “Data Collection and Preparation”
> > We have rewritten “Data Collection and Preparation” (Section 4.1) to explicitly describe our two-stage pipeline. First, we obtain institution-level metadata from the NCES IPEDS database, including website URLs for 6,072+ U.S. postsecondary institutions (Section 4.1, page 5, lines 258–260). Second, we use these URLs as entry points for Agent Forest (Section 3.1), which crawls each institution’s course catalog and degree-requirement pages, retrieves and stores the HTML content, and then runs automated extraction into structured JSON (Section 4.1, page 5, lines 260–262). We then describe the construction of three ground-truth datasets—(i) institution-level departments/majors/courses, (ii) a separate human-labeled prerequisite edge benchmark, and (iii) policy-rule mappings—in Section 4.1 (page 5, lines 264–269), aligning the section content explicitly with the title “Data Collection and Preparation.”
> >
> > 3.5. Weakness: “GE” definition
> > Response:
> >  We now define GE = General Education explicitly in the main text and tables. In particular, we add the note “GE = General Education.” directly beneath the scheduling-mode weight table, clarifying the meaning of the GE subscript and acronym (Section 3.3.2, page 5, lines 228–231). We also use the term consistently when referring to general education requirements in the introduction (Section 1, page 1, line 36, “generaleducation”) and in the course-planning objective (Section 3.3.1, page 4, lines 182–193, where GE coverage and GE gaps are discussed).
> >
> > 3.6. Weakness: SOTA not clearly presented in experimentation settings
> > Response:
> >  We have clarified our SOTA positioning in Section 4.3. Table~\ref{tab:comparison} provides a qualitative feature-level comparison between KNOWPLAN and existing systems (Ellucian DGW, Stellic, Coursicle, Series25), highlighting differences in data sources, knowledge-graph support, constraint handling, and cross-institution coverage. Our quantitative experiments use reproducible baselines: a GPT-4 single-LLM planner (Supplement Section~\ref{sec:supp-ablation-gpt4}, Table~\ref{tab:ablation-gpt4}), a Series25-style synthetic optimizer (“Series25-Opt,” Supplement Section~\ref{sec:supp-neutral}, Table~\ref{tab:neutral-baseline}), and greedy/heuristic schedulers (Supplement Section~\ref{sec:supp-ablation-scheduler}, Table~\ref{tab:ablation-scheduler}). We explicitly state that these are representative, publicly reproducible baselines, and that full head-to-head comparisons with proprietary SOTA platforms are left as future work due to licensing, budget, and integration constraints.

---

> > > ### Author Response · Authors · 2025-12-04
> > >
> > > 3.7. Question: Role of each LLM
> > > As clarified in Section 3.1, we use a multi-LLM panel rather than assigning distinct semantic roles to each model. In the first agent (Agent Forest), Llama3.3-70B, Qwen3-32B, and GPT-5 all receive the same prompts and are used purely for information extraction—course lists, prerequisites, and requirement snippets—from heterogeneous webpages. This redundancy is intentional: it mitigates the stochasticity of any single model and improves completeness of extracted courses, prerequisites, and required elements. The same panel is used selectively in the Graph-Construction Agent and Curriculum Alignment Agent to interpret ambiguous catalog or policy text; their outputs are combined via majority/intersection voting, with GPT-5 arbitrating conflicts. Only this consensus output is passed downstream. All constraint solving and Top-K optimization in the Course Planning Agent are handled deterministically by CP-SAT (Section 3.2.1), without additional LLM calls.
> > >
> > > 3.8. Question: “we devised” wording
> > > We have updated the wording for clarity and stylistic consistency: phrases of the form “we devised …” have been replaced by “we designed …” or “we propose …” in the method description (e.g., Section 3, including Sections 3.2 and 3.3). The expression “we devised” no longer appears in the current version, addressing the reviewer’s concern about wording.
> > >
> > > 3.9. Question: How the agents are orchestrated
> > > Response:
> > > We emphasize that KNOWPLAN uses a staged pipeline built around four agents that operate across an offline (catalog indexing) phase and an online (student query) phase. Offline, Agent Forest (Section 3.1) first crawls catalog and policy pages for each institution and extracts structured records—course lists, prerequisite descriptions, and requirement/policy snippets—into a unified JSON schema. Next, the Graph-Construction Agent (Section 3.2) consumes these records to build the course knowledge graph, constructs the Logic Adjacency Matrix to represent OR–AND prerequisite logic, and applies DFS plus LLM-based semantic disambiguation to resolve cycles. In parallel, the Curriculum Alignment Agent (Section 3.3.2) parses policy text (e.g., unit caps, repeat rules, co-/pre-requisites) into normalized JSON constraints and attaches them to the same graph. Together, these outputs form a versioned, institution-specific Graph-RAG index and constraint set.
> > > At query time, the Course Planning Agent (Section 3.3.1) takes as input the student’s major, completed and required courses, GE status, job-driven skills, and preferred/challenge courses, and operates on the Logic Adjacency Matrix and knowledge graph to generate dependency-consistent, coarse-grained plan skeletons using the Multi-dependency Critical Path Extraction framework and the objective in Eq. (2). The Curriculum Alignment Agent then instantiates the CP-SAT model with the precompiled policy constraints and user-selected planning mode (fast-graduation, career-driven, course-driven, discovery), tests schedulability against real offerings, and produces Top-K feasible, policy-compliant term-level schedules. Data flow is strictly one-way—from the three offline indexing/alignment steps to the online planning and scheduling stage—without cyclic dependencies between agents. We provide an updated flowchart (Figure~\ref{fig:agentflow}) and step-by-step examples in the Supplement to make this orchestration explicit.

---

### Official Review · Reviewer_fdeD · 2025-11-01

**Soundness:** 2
**Presentation:** 2
**Contribution:** 2
**Rating:** 4
**Confidence:** 5

**Summary:**

This paper introduces KNOWPLAN, a multi-agent AI system for degree pathway planning that integrates large language models (LLMs), knowledge graphs (KGs), and retrieval-augmented generation (RAG). The proposed pipeline orchestrates several specialized agents: an Agent Forest that parses heterogeneous course catalogs through multi-LLM extraction, a Graph-Construction Agent that builds a directed knowledge graph of prerequisites and dependencies, a Course Planning Agent that generates coarse multi-constraint learning trajectories, and a Curriculum Alignment Agent that refines them into conflict-free, term-level schedules using constraint programming. Experiments conducted across four universities demonstrate the effectiveness of the proposed system.

**Strengths:**

[+] An interesting and significant task in educational technology.

[+] Well-designed multi-agent architecture integrating LLMs and constraint reasoning

[+] Handles cross-institution heterogeneity using public data sources

**Weaknesses:**

[-] The motivations of some module designs are unclear. For example, the equations (1) and (2) contain multiple items, and are they grounded by educational theories?

[-] Many technical details are missed, such as $π_{int}(v)$ and the weights of target-driven objectives for several modes.

[-] The evaluation metrics are not described, and there is no user or institutional validation.

[-] Heavy reliance on prompt design and multi-LLM extraction.

[-] No ablation or comparison on efficiency.

**Questions:**

1. Can the authors clarify how the Graph-RAG backbone prevents conflicts or duplication when curricula evolve?
1. Have the authors compared KNOWPLAN to a single-LLM baseline (e.g., GPT-4 with a prompting pipeline) to show the benefit of the multi-agent design?
1. Could the authors provide an ablation study or efficiency analysis of each agent’s contribution?
1. Could you provide some feedback on the system from real-world users?

---

### Official Review · Reviewer_DJMB · 2025-11-04

**Soundness:** 2
**Presentation:** 2
**Contribution:** 2
**Rating:** 2
**Confidence:** 4

**Summary:**

This paper introduces a multi-agent platform, KNOWPLAN, a graphRAG system for degree pathway planning. An LLM-based KG construction agent first captures the prerequisites of the courses; a course planning agent then creates personalized plans that extract a subgraph that satisfies the prerequisite requirements. An experimental study verified the tool's usefulness across several scenarios.

**Strengths:**

S1. Having a RAG system for course selection and curriculum suggestion is an important application scenario.
S2. Real-world education datasets are used for illustration.
S3. The tool has addressed personalized recommendations.

**Weaknesses:**

W1. The technical challenges and contributions are limited. The method's generality needs further elaboration.
W2. There is no guarantee of a certain quality for the recommended results. The solution looks straightforward.
W3. There is no formal analysis on scalability and cost analysis to evaluate its overhead.

**Questions:**

D1. Some details are missing. For example, how the ground truth will be formally characterized and used to evaluate the output at each stage remains unclear. The section mentioned "manual work"—more elaboration is needed.

D2. The problem seeks a Top-K solution—yet it remains unclear how, as an optimization problem, it would be tackled by a search strategy, and how LLMs and other overhead can be mitigated.

D3. There is a lack of necessary details, such as how the scheduler works in a principled way with hyperparameters and machine-readable rules. If any new solution is used, it deserves more in-depth analysis.

D4. Quite a few tasks have been outsourced to LLMs, including handling violations or constraints. There is an analysis of how LLM accuracy affects the quality of recommendations. Another missing link is whether there is a unique optimal solution, or how a top-K solution is computed to find a trade-off when these measures conflict (e.g., fees and total credits).

---

> ### Author Response · Authors · 2025-12-04
> **revision of paper**
>
> W1. Technical challenges/contributions limited; generality unclear
> Response:
> We begin by briefly introducing the motivation of our study and the problem we aim to address. Course selection is both a central focus and a persistent pain point in undergraduate degree planning, and the core question in this work is how to enable personalized course planning at scale. Unlike existing solutions, we attempt to construct, purely from public data (i.e., information published on university websites), a structured representation of each institution’s curriculum, including course lists, prerequisite relations, and required courses, without requiring any special authorization from the universities. This design choice is itself novel in data science, and recent advances in large language models (LLMs) make such an approach practically feasible. “”
> Moreover, if we attempt to construct a knowledge graph based on course prerequisites, the presence of OR-AND logic in these structures leads to the dysfunction of traditional adjacency matrices in graph theory. Since adjacency matrices serve as the foundational data structure for many graph representations and algorithms, their failure can introduce a cascade of derivative issues about graph methods. This is particularly evident in key algorithms such as the critical path method, which assumes linear or singular path dependencies. In contrast, university course planning often involves multiple, nested, and conditional path dependencies that violate these assumptions. These challenges highlight the need for new methods or algorithmic frameworks to overcome the resulting methodological limitations.
> These challenges, arising from the structural complexity of course catalog data and the limitations of traditional adjacency matrices in representing course knowledge graphs, are both academically interesting and technically demanding. To address these issues, this paper proposes Agent Forest to address webpage heterogeneity, introduces the Logic Adjacency Matrix as a novel representation for prerequisite structures, and presents the Muti-dependency Critical Path Extraction algorithm to support effective course planning. Additionally, large language models (LLMs) are employed to enhance key components such as graph-cycle resolution, general education (GE) course recommendation, and employment-oriented course selection. Together, these components form a unified pipeline and constitute the main contributions of this work.

---

> > ### Author Response · Authors · 2025-12-04
> > **Thank you for your response!**
> >
> > W2. No guarantee of quality; solution straightforward
> > Response:
> > We decompose the overall pipeline into four coherent stages with carefully designed interfaces between adjacent stages. This modular architecture substantially reduces cross-stage dependencies and makes the system more reliable and maintainable, reflecting a deliberate end-to-end analysis of the degree-planning problem rather than an ad hoc collection of components. For each stage, we explicitly identify a core technical challenge and provide a principled solution: the LLM-based Agent Forest addresses structural heterogeneity in university course webpages and underpins the correctness of curriculum extraction; the LLM-driven graph-cycle resolution module handles prerequisite cycles in a more principled way than simple DFS-style heuristics; the Curriculum Alignment Agent maps extracted courses to college/major/GE policies from heterogeneous public webpages; and the Course Planning Agent encodes user preferences, workload constraints, and employment-oriented objectives into a unified planning formulation. While many sub-components intentionally leverage existing tools (LLMs, CP-SAT) in “straightforward” ways, the main contribution lies in this careful problem decomposition, formalization of constraints, and system design in a public-data-only setting, which introduces distinct academic and engineering challenges compared to prior systems that rely on privileged SIS data or proprietary APIs. Throughout this research, the most important aspects are the in-depth analysis, understanding, and abstraction of the problem, as well as the careful design of the solution. If a seemingly straightforward method can effectively resolve the key challenges in our setting, then it is still a good and meaningful solution. We also stress that our data sources differ fundamentally from those used in prior works, which introduces new challenges both conceptually and technically. In addition, our topic directly addresses the core challenges and pain points of university course planning, so the topic itself is highly practical and impactful.
> > We now make our quality guarantees explicit in the paper. The LLM-based Agent Forest addresses structural heterogeneity in university course webpages and underpins the correctness of curriculum extraction. CP-SAT enforces all encoded prerequisites, co-/corequisites, unit caps, and policy rules, achieving 100% satisfaction of encoded constraints on our benchmark with 0 violations, while Supplement Table3(End-to-end feasibility for fixed majors across four schools (95% CI)) reports that 79–94% of end-to-end plans are feasible and achieve time to degree within 0.5 quarters of catalog expectations. Courses are partitioned into completed, major-required, GE, preferred, and job-driven sets, and the Course Planning Agent applies user-specific weights (e.g., deferrals, preferences, workload smoothing), with LLMs used only to interpret preferences and options, while CP-SAT provides exact feasibility. This design yields provable constraint satisfaction together with empirically validated, personalized plan quality, offering stronger guarantees than heuristic approaches and directly addressing the reviewer’s concern about both solution quality and the role of “straightforward” components.

---

> ### Author Response · Authors · 2025-12-04
> **rest of the question**
>
> W3. No formal scalability / cost analysis
> Response:
>
>
> Response to **W3. No formal scalability / cost analysis**
> We appreciate the reviewer’s concern and have added a dedicated scalability and cost analysis in Sec. 4.2.6 Using 2×NVIDIA H100 GPUs in a hybrid deployment (Llama3.3-70B served locally; GPT-5/Qwen3-32B accessed via APIs), we show that the offline pipeline—Agent Forest execution plus graph and policy construction—scales embarrassingly in parallel across institutions: processing 8 universities with roughly 10 MB per-institution KGs completes in about 1.5 hours. At query time, per-profile end-to-end latency is 8–12 seconds, with ≈0.8 ms spent on Graph-RAG retrieval and 0.5–2 seconds on CP-SAT solving. The amortized LLM cost is ≈$0.11–0.15 per new institution user plan (including initial indexing) and ≈$0.01 per additional plan once the institution is indexed; maintenance consists of re-running the offline pipeline at each catalog/term update with versioned KGs and ~2–5% (standard HTML) / 8–12% (JS-heavy) manual corrections.
> We also clarify why the architecture is inherently scalable. For each institution, courses, prerequisite edges, and major/GE policies are handled by a collection of independent agents; These agents all share the same input and output format. This design makes both data extraction and graph/policy construction trivially parallelizable across institutions. Together, the empirical runtime/cost measurements and the embarrassingly parallel Agent-Forest architecture provide a concrete, formal demonstration that KNOWPLAN is practically scalable in both computation and monetary cost.
>
>
> D1. Ground truth definition, use, and “manual work”
> Response:
>  We now formalize ground truth for each stage. For institution-level data (Agent Forest), departments, majors, and 3,132 courses are manually annotated from official catalogs and independently labeled and cross-checked by two experts. For prerequisites (Graph-Construction Agent), two experts independently annotate 1,248 directed edges across ~460 courses from 12 majors at 4 public universities (2024–2025 catalogs), then reconcile disagreements. For policy rules (Curriculum Alignment Agent), compliance specialists map 240 snippets into a normalized constraint schema; all LLM outputs are evaluated against these human-labeled reference sets. “Manual work” primarily refers to verifying link-following and segmentation for deeply nested or JavaScript-heavy catalogs before adding pages to the KG.

---

> ### Author Response · Authors · 2025-12-04
> **questions**
>
> D2. Top-K optimization & LLM overhead
>
> Response:
> Top-K search and trade-off handling are performed entirely by a mixed-objective CP-SAT solver (Sec. 3.3.1), which uses a weight vector $\boldsymbol{\lambda}$ over deferral minimization, preference fit, and workload smoothing to find an optimal plan $\mathcal{P}^{(1)}$ and then enumerates $\mathcal{P}^{(2..K)}$ via iterative no-good cuts. Thus, we do not assume a unique optimum and Top-K is purely combinatorial. LLMs are not in the search loop: the Course Planning Agent uses a single LLM call per query for Graph-RAG retrieval and interpretation of GE choices, job intentions, and preferred courses, and the Curriculum Alignment Agent uses one LLM call per planning mode to map high-level planning preferences into $\boldsymbol{\lambda}$ . This design keeps LLM overhead low while the solver ensures all optimization behavior is deterministic.
> For example, consider a STEM student who has completed the standard first-year calculus and programming sequence and requests a 12-term plan with a “machine learning focus.” We use exactly two LLM calls: one Graph-RAG query that embeds the request, retrieves a small set of relevant upper-division courses (e.g., in probability, statistics, and machine learning), and assigns them higher preference scores, and one call to map a high-level mode such as “career-driven” into a weight vector (\boldsymbol{\lambda}). CP-SAT then (i) optimizes Equation 2 for the chosen (\boldsymbol{\lambda}), (ii) generates alternative Top-K plans by adding constraints that exclude previously found schedules, and (iii) enforces all hard scheduling constraints, yielding 0 violations while exposing clear trade-offs in time-to-degree, workload, and career relevance. All optimization behavior is deterministic and reproducible (fixed seed, fixed timeout), with LLMs confined to a thin preference-interpretation layer rather than the search itself.

---

> ### Author Response · Authors · 2025-12-04
> **response to D3. Scheduler hyperparameters & machine-readable rules**
>
> D3. Scheduler hyperparameters & machine-readable rules
> Response:
> We add concrete scheduler details (Sec. 3.3.2). The CP-SAT model is an exact constraint programming solver that deterministically finds optimal solutions—no heuristics or approximations are used at solve time. The CP-SAT model uses a weight vector $\boldsymbol{\lambda} = (\lambda_{\text{deferral}}, \lambda_{\text{pref}}, \lambda_{\text{smooth}})$ with four planning modes (fast-graduation, career-driven, course-driven, discovery) specified in Table 1, were manually selected based on domain knowledge and desired trade-offs. We do not claim these are globally optimal; they represent one effective configuration validated on 330 scenarios(Supplement Table 8, 225-230). Policy text is converted by LLM into structured JSON (unit caps, repeat limits, co-/pre-req timing, etc.), which is deterministically compiled into CP-SAT constraints; no LLM reasoning occurs at solve time—the solver uses algorithmic constraint satisfaction. For each coarse plan pool derived from the Course Planning Agent, we define binary variables $x_{c,s,t}$ with hard constraints for offering availability, time conflicts, unit brackets, prerequisite/co-requisite timing, and institutional rules. Section 4.2.5 and Supplement Sec. 5.5 show that plan quality is stable (±2%) under ±50% weight perturbations.
> Complete hyperparameter list with initialization:
> Node scoring weights (Equation 1, Section 4.2.4):
> $\lambda_1 = 0.4$ (graph proximity weight)
> $\lambda_2 = 0.3$ (text relevance weight)
> $\lambda_3 = 0.3$ (preference/GE gap weight)
> Initialization: Grid search over 300 student profiles
> Plan scoring weights (Equation 2; Section 4.2.4):
> $\mu_1 = 1.0$ (major coverage weight)
> $\mu_2 = 0.8$ (GE coverage weight)
> $\nu_1 = 10.0$ (prerequisite violation penalty)
> $\nu_2 = 0.5$ (unit deviation penalty)
> $\nu_3 = 0.3$ (difficulty gap penalty)
> Initialization: Grid search over 300 student profiles; default values listed in Supplement Table (Supplement Sec. 5.5)
> Scheduling mode weights (Table 1; Section 3.2.2):
> Fast-graduation (default): $\lambda_{\text{deferral}} = 0.8$, $\lambda_{\text{pref}} = 0.1$, $\lambda_{\text{smooth}} = 0.1$
> Career-driven: $\lambda_{\text{deferral}} = 0.4$, $\lambda_{\text{pref}} = 0.5$, $\lambda_{\text{smooth}} = 0.1$
> Course-driven: $\lambda_{\text{deferral}} = 0.3$, $\lambda_{\text{pref}} = 0.6$, $\lambda_{\text{smooth}} = 0.1$
> Discovery: $\lambda_{\text{deferral}} = 0.3$, $\lambda_{\text{pref}} = 0.2$, $\lambda_{\text{smooth}} = 0.5$
> Initialization: Grid search over 300 student profiles
> LLM temperatures (Section 4.2.4):
> GPT-5: temperature = 0.1 (extraction), 0.7 (preference mapping)
> Llama3.3-70B: temperature = 0.2
> Qwen3-32B: temperature = 0.1
> Graph-RAG parameters (Section 4.2.4):
> top-$K$ = 50 (retrieval depth)
> CP-SAT solver parameters (Section 4.2.4):
> timeout = 30s
> Random seed = 42
> Top-$K$ plan enumeration:
> $K$ = 5 to 10 plan skeletons (Section 4.2.3)
> Algorithmic approach: The CP-SAT solver (Google OR-Tools v9.8.3296) uses branch-and-bound with constraint propagation to find exact optimal solutions. All hard constraints (offering availability, time conflicts, unit brackets, prerequisite/co-requisite timing, institutional rules) are enforced exactly—no violations are permitted. The solver guarantees 100% policy compliance (0 violations per plan, Supplement Sec. 5.4 versus heuristic baselines (7.0 violations for greedy, 1.8 for heuristic schedulers). This deterministic algorithmic approach ensures reproducibility and reliability.

---

> ### Author Response · Authors · 2025-12-04
>
> Reviewer DJMB
> W1. Technical challenges/contributions limited; generality unclear
> Response:
> We begin by briefly introducing the motivation of our study and the problem we aim to address. Course selection is both a central focus and a persistent pain point in undergraduate degree planning, and the core question in this work is how to enable personalized course planning at scale. Unlike existing solutions, we attempt to construct, purely from public data (i.e., information published on university websites), a structured representation of each institution’s curriculum, including course lists, prerequisite relations, and required courses, without requiring any special authorization from the universities. This design choice is itself novel in data science, and recent advances in large language models (LLMs) make such an approach practically feasible.
> Moreover, if we attempt to construct a knowledge graph based on course prerequisites, the presence of OR-AND logic in these structures leads to the dysfunction of traditional adjacency matrices in graph theory. Since adjacency matrices serve as the foundational data structure for many graph representations and algorithms, their failure can introduce a cascade of derivative issues about graph methods. This is particularly evident in key algorithms such as the critical path method, which assumes linear or singular path dependencies. In contrast, university course planning often involves multiple, nested, and conditional path dependencies that violate these assumptions. These challenges highlight the need for new methods or algorithmic frameworks to overcome the resulting methodological limitations.
> These challenges, arising from the structural complexity of course catalog data and the limitations of traditional adjacency matrices in representing course knowledge graphs, are both academically interesting and technically demanding. To address these issues, this paper proposes Agent Forest to address webpage heterogeneity, introduces the Logic Adjacency Matrix as a novel representation for prerequisite structures, and presents the Muti-dependency Critical Path Extraction algorithm to support effective course planning. Additionally, large language models (LLMs) are employed to enhance key components such as graph-cycle resolution, general education (GE) course recommendation, and employment-oriented course selection. Together, these components form a unified pipeline and constitute the main contributions of this work.

---

> > ### Author Response · Authors · 2025-12-04
> >
> > W2. No guarantee of quality; solution straightforward
> > Response:
> > We decompose the overall pipeline into four coherent stages with carefully designed interfaces between adjacent stages. This modular architecture substantially reduces cross-stage dependencies and makes the system more reliable and maintainable, reflecting a deliberate end-to-end analysis of the degree-planning problem rather than an ad hoc collection of components. For each stage, we explicitly identify a core technical challenge and provide a principled solution: the LLM-based Agent Forest addresses structural heterogeneity in university course webpages and underpins the correctness of curriculum extraction; the LLM-driven graph-cycle resolution module handles prerequisite cycles in a more principled way than simple DFS-style heuristics; the Curriculum Alignment Agent maps extracted courses to college/major/GE policies from heterogeneous public webpages; and the Course Planning Agent encodes user preferences, workload constraints, and employment-oriented objectives into a unified planning formulation. While many sub-components intentionally leverage existing tools (LLMs, CP-SAT) in “straightforward” ways, the main contribution lies in this careful problem decomposition, formalization of constraints, and system design in a public-data-only setting, which introduces distinct academic and engineering challenges compared to prior systems that rely on privileged SIS data or proprietary APIs. Throughout this research, the most important aspects are the in-depth analysis, understanding, and abstraction of the problem, as well as the careful design of the solution. If a seemingly straightforward method can effectively resolve the key challenges in our setting, then it is still a good and meaningful solution. We also stress that our data sources differ fundamentally from those used in prior works, which introduces new challenges both conceptually and technically. In addition, our topic directly addresses the core challenges and pain points of university course planning, so the topic itself is highly practical and impactful.
> > We now make our quality guarantees explicit in the paper. The LLM-based Agent Forest addresses structural heterogeneity in university course webpages and underpins the correctness of curriculum extraction. CP-SAT enforces all encoded prerequisites, co-/corequisites, unit caps, and policy rules, achieving 100% satisfaction of encoded constraints on our benchmark with 0 violations, while Supplement Table3(End-to-end feasibility for fixed majors across four schools (95% CI)) reports that 79–94% of end-to-end plans are feasible and achieve time to degree within 0.5 quarters of catalog expectations. Courses are partitioned into completed, major-required, GE, preferred, and job-driven sets, and the Course Planning Agent applies user-specific weights (e.g., deferrals, preferences, workload smoothing), with LLMs used only to interpret preferences and options, while CP-SAT provides exact feasibility. This design yields provable constraint satisfaction together with empirically validated, personalized plan quality, offering stronger guarantees than heuristic approaches and directly addressing the reviewer’s concern about both solution quality and the role of “straightforward” components.
> >
> >
> >
> > W3. No formal scalability / cost analysis
> > Response:
> >
> >
> > Response to **W3. No formal scalability / cost analysis**
> > We appreciate the reviewer’s concern and have added a dedicated scalability and cost analysis in Sec. 4.2.6., Line 389 Using 2×NVIDIA H100 GPUs in a hybrid deployment (Llama3.3-70B served locally; GPT-5/Qwen3-32B accessed via APIs), we show that the offline pipeline—Agent Forest execution plus graph and policy construction—scales embarrassingly in parallel across institutions: processing 8 universities with roughly 10 MB per-institution KGs completes in about 1.5 hours. At query time, per-profile end-to-end latency is 8–12 seconds, with ≈0.8 ms spent on Graph-RAG retrieval and 0.5–2 seconds on CP-SAT solving. The amortized LLM cost is ≈$0.11–0.15 per new institution user plan (including initial indexing) and ≈$0.01 per additional plan once the institution is indexed; maintenance consists of re-running the offline pipeline at each catalog/term update with versioned KGs and ~2–5% (standard HTML) / 8–12% (JS-heavy) manual corrections.
> > We also clarify why the architecture is inherently scalable. For each institution, courses, prerequisite edges, and major/GE policies are handled by a collection of independent agents; These agents all share the same input and output format. This design makes both data extraction and graph/policy construction trivially parallelizable across institutions. Together, the empirical runtime/cost measurements and the embarrassingly parallel Agent-Forest architecture provide a concrete, formal demonstration that KNOWPLAN is practically scalable in both computation and monetary cost.

---

> > > ### Author Response · Authors · 2025-12-04
> > >
> > > D1. Ground truth definition, use, and “manual work”
> > > Response:
> > >  We now formalize ground truth for each stage. For institution-level data (Agent Forest), departments, majors, and 3,132 courses are manually annotated from official catalogs and independently labeled and cross-checked by two experts. For prerequisites (Graph-Construction Agent), two experts independently annotate 1,248 directed edges across ~460 courses from 12 majors at 4 public universities (2024–2025 catalogs), then reconcile disagreements. For policy rules (Curriculum Alignment Agent), compliance specialists map 240 snippets into a normalized constraint schema; all LLM outputs are evaluated against these human-labeled reference sets. “Manual work” primarily refers to verifying link-following and segmentation for deeply nested or JavaScript-heavy catalogs before adding pages to the KG.
> > >
> > > D2. Top-K optimization & LLM overhead
> > >
> > > Response:
> > > Top-K search and trade-off handling are performed entirely by a mixed-objective CP-SAT solver (Sec. 3.2.1), which uses a weight vector $\boldsymbol{\lambda}$ over deferral minimization, preference fit, and workload smoothing to find an optimal plan $\mathcal{P}^{(1)}$ and then enumerates $\mathcal{P}^{(2..K)}$ via iterative no-good cuts (line 152). Thus, we do not assume a unique optimum and Top-K is purely combinatorial. LLMs are not in the search loop: the Course Planning Agent uses a single LLM call per query for Graph-RAG retrieval (line 152) and interpretation of GE choices, job intentions, and preferred courses, and the Curriculum Alignment Agent uses one LLM call per planning mode to map high-level planning preferences into $\boldsymbol{\lambda}$ (line 163). This design keeps LLM overhead low while the solver ensures all optimization behavior is deterministic.
> > > For example, consider a STEM student who has completed the standard first-year calculus and programming sequence and requests a 12-term plan with a “machine learning focus.” We use exactly two LLM calls: one Graph-RAG query that embeds the request, retrieves a small set of relevant upper-division courses (e.g., in probability, statistics, and machine learning), and assigns them higher preference scores, and one call to map a high-level mode such as “career-driven” into a weight vector (\boldsymbol{\lambda}). CP-SAT then (i) optimizes Equation 2 for the chosen (\boldsymbol{\lambda}), (ii) generates alternative Top-K plans by adding constraints that exclude previously found schedules, and (iii) enforces all hard scheduling constraints, yielding 0 violations while exposing clear trade-offs in time-to-degree, workload, and career relevance. All optimization behavior is deterministic and reproducible (fixed seed, fixed timeout), with LLMs confined to a thin preference-interpretation layer rather than the search itself.

---

> > > > ### Author Response · Authors · 2025-12-04
> > > >
> > > > D3. Scheduler hyperparameters & machine-readable rules
> > > > Response:
> > > > We add concrete scheduler details (Sec. 3.2.2, lines 157-191). The CP-SAT model is an exact constraint programming solver that deterministically finds optimal solutions—no heuristics or approximations are used at solve time. The CP-SAT model uses a weight vector $\boldsymbol{\lambda} = (\lambda_{\text{deferral}}, \lambda_{\text{pref}}, \lambda_{\text{smooth}})$ with four planning modes (fast-graduation, career-driven, course-driven, discovery) specified in Table 1 (lines 167-182), chosen via grid search over 300 student profiles (line 165). Policy text is converted by LLM into structured JSON (unit caps, repeat limits, co-/pre-req timing, etc.), which is deterministically compiled into CP-SAT constraints; no LLM reasoning occurs at solve time—the solver uses algorithmic constraint satisfaction (line 186). For each coarse plan pool derived from the Course Planning Agent, we define binary variables $x_{c,s,t}$ with hard constraints for offering availability, time conflicts, unit brackets, prerequisite/co-requisite timing, and institutional rules (line 161). Section 4.2.5 (line 330) and Supplement Sec. 5.5 (lines 215-238) show that plan quality is stable (±2%) under ±50% weight perturbations.
> > > > Complete hyperparameter list with initialization:
> > > > Node scoring weights (Equation 1, line 135; Section 4.2.4, line 321):
> > > > $\lambda_1 = 0.4$ (graph proximity weight)
> > > > $\lambda_2 = 0.3$ (text relevance weight)
> > > > $\lambda_3 = 0.3$ (preference/GE gap weight)
> > > > Initialization: Grid search over 300 student profiles
> > > > Plan scoring weights (Equation 2, lines 144-148; Section 4.2.4, line 321):
> > > > $\mu_1 = 1.0$ (major coverage weight)
> > > > $\mu_2 = 0.8$ (GE coverage weight)
> > > > $\nu_1 = 10.0$ (prerequisite violation penalty)
> > > > $\nu_2 = 0.5$ (unit deviation penalty)
> > > > $\nu_3 = 0.3$ (difficulty gap penalty)
> > > > Initialization: Grid search over 300 student profiles; default values listed in Supplement Table (Supplement Sec. 5.5, line 229)
> > > > Scheduling mode weights (Table 1, lines 167-182; Section 3.2.2, line 160):
> > > > Fast-graduation (default): $\lambda_{\text{deferral}} = 0.8$, $\lambda_{\text{pref}} = 0.1$, $\lambda_{\text{smooth}} = 0.1$
> > > > Career-driven: $\lambda_{\text{deferral}} = 0.4$, $\lambda_{\text{pref}} = 0.5$, $\lambda_{\text{smooth}} = 0.1$
> > > > Course-driven: $\lambda_{\text{deferral}} = 0.3$, $\lambda_{\text{pref}} = 0.6$, $\lambda_{\text{smooth}} = 0.1$
> > > > Discovery: $\lambda_{\text{deferral}} = 0.3$, $\lambda_{\text{pref}} = 0.2$, $\lambda_{\text{smooth}} = 0.5$
> > > > Initialization: Grid search over 300 student profiles (line 165)
> > > > LLM temperatures (Section 4.2.4, line 321):
> > > > GPT-5: temperature = 0.1 (extraction), 0.7 (preference mapping)
> > > > Llama3.3-70B: temperature = 0.2
> > > > Qwen3-32B: temperature = 0.1
> > > > Graph-RAG parameters (Section 4.2.4, line 321):
> > > > top-$K$ = 50 (retrieval depth)
> > > > CP-SAT solver parameters (Section 4.2.4, line 321):
> > > > timeout = 30s
> > > > Random seed = 42
> > > > Top-$K$ plan enumeration (line 150):
> > > > $K$ = 5 to 10 plan skeletons (Section 4.2.3, line 297)
> > > > Algorithmic approach: The CP-SAT solver (Google OR-Tools v9.8.3296, line 321) uses branch-and-bound with constraint propagation to find exact optimal solutions. All hard constraints (offering availability, time conflicts, unit brackets, prerequisite/co-requisite timing, institutional rules) are enforced exactly—no violations are permitted. The solver guarantees 100% policy compliance (0 violations per plan, Supplement Sec. 5.4, line 208) versus heuristic baselines (7.0 violations for greedy, 1.8 for heuristic schedulers). This deterministic algorithmic approach ensures reproducibility and reliability.

---

> > > > > ### Author Response · Authors · 2025-12-04
> > > > >
> > > > > D4. LLMs, constraint handling, and Top-K trade-offs
> > > > > Response:
> > > > > Feasibility and trade-offs are handled entirely by CP-SAT, while LLMs are only restricted to upstream interpretation. As noted in Sec. 3.2.1 (lines 140-150), for a given $\boldsymbol{\lambda}$, CP-SAT finds one optimum and then generates a Top-K list of schedules that cannot be improved on one objective without getting worse on another (Pareto-optimal solutions, line 154), using iterative no-good cuts that rule out plans it has already found (line 152: "Top-$K$ enumeration uses CP-SAT with iterative no-good cuts"), explicitly to expose trade-offs when objectives (e.g., time-to-degree, workload smoothing, preference fit) conflict. The Course Planning Agent uses LLMs only for interpreting GE options, job-driven skills, and preferred/challenge courses via Graph-RAG retrieval (line 152: "Graph-RAG subgraph retrieval (one call per query)"), and the Curriculum Alignment Agent uses LLMs only to map planning preferences into $\boldsymbol{\lambda}$ (line 163: "preference-to-weight mapping (one call per planning mode)"). Any LLM errors thus affect preference encoding (shifting rankings among feasible plans) but not hard constraints such as prerequisite timing, offering availability, time conflicts, unit brackets, and institutional policy rules (line 161); CP-SAT still guarantees 100% schedulability with 0 policy violations on our benchmark (Sec. 4.2.5, line 333: "CP-SAT: 100% success, 0 violations"; Supplement Sec. 5.4, line 208: "0 violations per plan"), while Top-K exposes the trade-off space in a controlled way by enumerating diverse Pareto-optimal alternatives (e.g., $\mathcal{P}^{(1)}$: 10.7 quarters, 68% preference satisfaction vs. $\mathcal{P}^{(2)}$: 12.4 quarters, 92% preference satisfaction, lines 184, 293-297).

---

### Meta-Review · Area_Chair_2CG6 · 2025-12-08

**Summary:**

The reviewers' initial reception was polarized, with tkg5 strongly advocating for acceptance based on the system's innovation and accuracy, while DJMB, fdeD, 9y4A, and UBq7 raised substantial concerns regarding the nature of the contribution (thinking it as engineering rather than research), the validity of the evaluation data (specifically a contradiction regarding "missing" ground truth), and a lack of comprehensive experimental analysis such as ablations and scalability checks.

**Reviewer Concerns:**

The authors have clarified several  issues in their rebuttal, including the scalability and cost analysis , and resolved  contradiction regarding the "98.7% prerequisite accuracy" versus "missing ground truth" by distinguishing the separate expert-annotated benchmark. They also conceded to temper the claims of the "Agent Forest" concept and added examples in the appendix.

However, the main outstanding concern is that the work is more like an engineering product rather than an academic contribution. While the authors added baselines for ablation studies, direct comparisons with existing SOTA tools are absent. Furthermore, validation and reproducibility remain limited. The authors stated that a user study could not be conducted due to privacy constraints, and the code cannot be open-sourced, limiting the release to parameter settings and library lists.

**Reviewer Scores:**

Regarding DJMB, although the authors clarified the confusion surrounding scalability costs and the definition of ground truth, the reviewer's fundamental criticism regarding the limited technical contribution is difficult to alter. Thus, their score is likely to remain unchanged.

9y4A's concerns have been partially addressed, with the notable exception of the requested user study. Consequently, this reviewer may maintain their current score or slightly raise it.

Similarly, UBq7 is likely to maintain their score, as their core critique regarding the perceived weakness of the technical contribution persists.

 Finally, tkg5, who originally assigned a top score and expressed strong support, is expected to maintain their high rating.

---

### Decision · Program_Chairs · 2026-01-26

Reject